# NOISY-PAIR ROBUST REPRESENTATION ALIGNMENT FOR POSITIVE-UNLABELED LEARNING

**Hengwei Zhao** [1*]     **Zhengzhong Tu** [1]     **Zhuo Zheng** [2]     **Wei Wang** [3,4]     **Junjue Wang** [4]
**Rusty Feagin** [1]     **Wenzhe Jiao** [1*]
[1] Texas A&M University, College Station, USA     [2] Stanford University, Stanford, USA
[3] RIKEN, Tokyo, Japan     [4] The University of Tokyo, Chiba, Japan
{hengwei.zhao,wenzhe.jiao}@ag.tamu.edu

## ABSTRACT

Positive-Unlabeled (PU) learning aims to train a binary classifier (positive *vs.* negative) where only limited positive data and abundant unlabeled data are available. While widely applicable, state-of-the-art PU learning methods substantially underperform their supervised counterparts on complex datasets, especially without auxiliary negatives or pre-estimated parameters (*e.g.*, a 14.26% gap on CIFAR-100 dataset). We identify the primary bottleneck as the challenge of learning discriminative representations under unreliable supervision. To tackle this challenge, we propose *NcPU*, a non-contrastive PU learning framework that requires no auxiliary information. *NcPU* combines a noisy-pair robust supervised non-contrastive loss (NoiSNCL), which aligns intra-class representations despite unreliable supervision, with a phantom label disambiguation (PLD) scheme that supplies conservative negative supervision via regret-based label updates. Theoretically, NoiSNCL and PLD can iteratively benefit each other from the perspective of the Expectation-Maximization framework. Empirically, extensive experiments demonstrate that: (1) NoiSNCL enables simple PU methods to achieve competitive performance; and (2) *NcPU* achieves substantial improvements over state-of-the-art PU methods across diverse datasets, including challenging datasets on post-disaster building damage mapping, highlighting its promise for real-world applications. Code: https://github.com/Hengwei-Zhao96/NcPU.

## 1 INTRODUCTION

Positive-Unlabeled (PU) learning aims to train a binary classifier with limited labeled positive data and a large pool of unlabeled data (Zhao et al., 2023a; Long et al., 2024; Du Plessis et al., 2015), which is well-suited for many real-world applications such as product recommendation (Hsieh et al., 2015), medical diagnosis (Yuan et al., 2025), and remote sensing applications (Zhao et al., 2022). For example, a real-world case arises in the context of humanitarian assistance and disaster response (HADR), where mapping the spatial distribution of damaged buildings from remote sensing imagery remains highly challenging. Shortly after a disaster, specialists are typically able to annotate only a subset of damaged buildings (positive samples) (Xia et al., 2021), while a large amount of the remaining building data remains unlabeled. These unlabeled data inevitably contain both damaged and undamaged (negative) structures, and such ambiguity significantly impedes the effective training of classification models.

Numerous methods for PU learning have been developed over the past decades, and the core challenge in PU learning lies in deriving the reliable binary classification supervision using only PU data. Early studies focused on selecting reliable negative samples, and then training a binary classifier with positive and reliable negatives (Gong et al., 2018; Garg et al., 2021; Wang et al., 2023a). However, the performance of these methods heavily depends on the pseudo-labels of the selected samples. More recent research tries to estimate the reliable binary classification supervision using all PU data, and achieving competitive or state-of-the-art performance (Wilton et al., 2022; Jiang et al., 2023; Zhao et al., 2023a; Long et al., 2024; Yuan et al., 2025). Nevertheless, many of these

---

[*]Corresponding author.

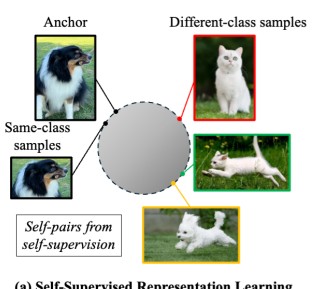 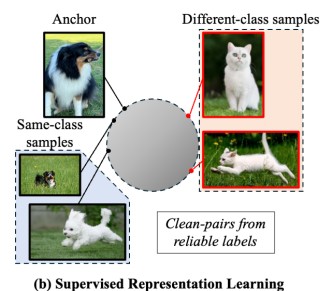 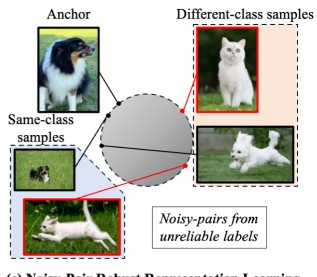

**(a) Self-Supervised Representation Learning** | **(b) Supervised Representation Learning** | **(c) Noisy-Pair Robust Representation Learning**

Figure 1: **Illustration of different representation learning methods.** Representation learning can acquire discriminative representations either by pulling same-class samples closer to the anchor and pushing different-class samples apart (contrastive representation learning), or by only pulling same-class samples closer to the anchor (non-contrastive representation learning). (a) Self-supervised representation learning: same-class pairs from augmented anchor. (b) Supervised representation learning: same-class pairs from reliable labels. (c) Noisy-pair robust representation learning: same-class pairs from unreliable labels.

methods rely on auxiliary negative validation data or a pre-estimated parameters (such as the class prior $\pi_p$, which denotes the proportion of positive samples within the unlabeled data) to facilitate the estimation of binary classification supervision.

While previous methods have shown promising results, learning discriminative representations from complex PU data remains a significant challenge, causing the best-performing methods to fall short compared to supervised counterparts. As shown in Figure 2, the features generated by PU methods, such as LaGAM (Long et al., 2024) and HolisticPU (Wang et al., 2023a), show substantial overlap between positive and negative distributions, unlike the clearly separable features obtained with supervised learning. Additional t-SNE visualizations are provided in the Appendix A. In other words, the binary classification supervision derived from PU data in these methods is insufficient to ensure the acquisition of discriminative representations, which adversely affects model performance.

Inspired by recent advances in self-supervised/supervised representation learning (He et al., 2020; Grill et al., 2020; Khosla et al., 2020; Lu et al., 2023), this paper proposes a novel non-contrastive PU learning framework (*NcPU*) to learn more discriminative representations. When representation learning meets PU learning, the challenge of noisy-pair robust representation learning (Figure 1) arises: unreliable supervision inevitably introduces incorrect pairwise relations (*i.e.*, noisy pairs), which hinder the effective learning of discriminative representations. To address this, noisy-pair robust supervised non-contrastive loss (NoiSNCL) is proposed to align intra-class representation while tolerating noisy pairs from the perspective of gradients. Moreover, based on the discriminative representations learned by NoiSNCL, the phantom label disambiguation (PLD) is proposed, which refines supervision through regret-based label updating. Theoretically, *NcPU* can be interpreted under the Expectation-Maximization (EM) framework, with pseudo-

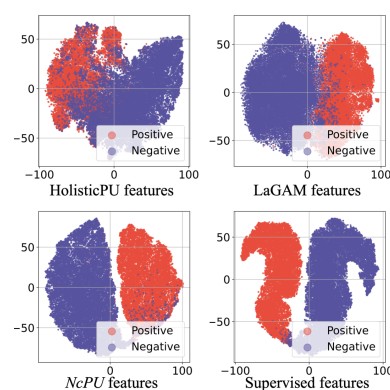

Figure 2: t-SNE visualizations of the representations learned by PU methods on CIFAR-10 training dataset.

label assignment serving as the E-step and NoiSNCL minimization as the M-step (cluster tightening), iteratively updated during training. Remarkably, NoiSNCL enables simple PU methods to achieve highly competitive performance. The contributions of this paper are summarized as follows:

- **Methodology**. *NcPU* is proposed to obtain discriminative representations through noisy-pair robust intra-class representations alignment, which consists of NoiSNCL and PLD modules. Notably, *NcPU* works well without requiring auxiliary negatives or pre-estimated parameters.
- **Theories**. Gradient analysis demonstrates that noisy pairs dominate the optimization process in representation learning. Furthermore, the proposed NoiSNCL and PLD modules can be theoretically justified to iteratively benefit each other from the perspective of the EM framework.

- **Experiments**. Extensive experiments demonstrate that (1) NoiSNCL enables simple PU methods to achieve competitive performance; and (2) *NcPU* outperforms state-of-the-art methods, achieving performance comparable to its supervised counterpart. Moreover, results on post-disaster building damage mapping tasks highlight the broad applicability of *NcPU*.

## 2 PROBLEM SETTING

Let $\boldsymbol{x}_i \in \mathbb{R}^d$ denote a sample and $y_i$ denote its label. Here, $y_i = 0$ indicates that $\boldsymbol{x}_i$ belongs to the positive class, and $y_i = 1$ implies that $\boldsymbol{x}_i$ belongs to the negative class. The objective of PU learning is to learn a binary classifier $f(\boldsymbol{x}_i) : \mathbb{R}^d \to [0, 1]^2$ using the training dataset $\mathcal{D} = \mathcal{P} \cup \mathcal{U}$:

$$\mathcal{P} = \{(\boldsymbol{x}_i, y_i = 0)\}_{i=1}^{n_p} \sim \mathbb{P}_p = P(\boldsymbol{x}|y = 0), \tag{1}$$

$$\mathcal{U} = \{\boldsymbol{x}_i\}_{i=1}^{n_u} \sim \mathbb{P}_u = \pi_p P(\boldsymbol{x}|y = 0) + (1 - \pi_p) P(\boldsymbol{x}|y = 1), \tag{2}$$

where $n_p$ and $n_u$ denote the numbers of positive and unlabeled samples, respectively, and $\mathbb{P}_p$ and $\mathbb{P}_u$ denote the marginal distributions of positive and unlabeled data, respectively. Besides, $\pi_p = P(y = 0)$ is the class prior.

The challenge of PU learning is to derive reliable binary classification supervision from the PU data. Let a pseudo target $\boldsymbol{s}_i \in [0, 1]^2$ be assigned to a sample $\boldsymbol{x}_i$, $\boldsymbol{s}_i$ is expected to become more accurate as it is updated during training without auxiliary information. If the $\boldsymbol{s}_i$ is obtainable, the classifier can be optimized by minimizing the following label-disambiguation cross-entropy loss (LDCE):

$$\mathcal{L}_c(f(\boldsymbol{x}_i), \boldsymbol{s}_i) = -\boldsymbol{s}_i^\top \log f(\boldsymbol{x}_i), \tag{3}$$

where $f(\boldsymbol{x}_i)$ is the softmax output of the classifier. The sample index $i$ will be omitted if the context is clear in the following.

Intuitively, if ideal representations are available, the labels of unlabeled data can be accurately recovered by exploiting the available information: the similarity between data points in the representation space, and the weak supervision inherent in PU data. For example, an unlabeled sample would be inferred as belonging to the negative class if, in the ideal representation space, it lies close to a cluster corresponding to the negative class. However, this reliance on representations gives rise to a non-trivial dilemma: the inherent weak supervision adversely affects the process of representation learning (as illustrated in Figure 2), while the quality of the learned representations, in turn, constrains the acquisition of accurate pseudo targets. This dilemma is mitigated in *NcPU* through intra-class representation alignment, which is described in the following section.

## 3 NOISY-PAIR ROBUST NON-CONTRASTIVE PU LEARNING FRAMEWORK

The *NcPU* framework (Figure 3) is designed to obtain more discriminative representations by aligning intra-class representations. It consists of two key components: NoiSNCL and PLD. The two components work collaboratively: NoiSNCL produces discriminative representations under unreliable supervision, which in turn benefits subsequent label disambiguation. Conversely, PLD provides more reliable supervision, thereby facilitating the non-contrastive representation learning module in acquiring more discriminative representations. Notably, *NcPU* works well without requiring additional negative samples or pre-estimated parameters. A more detailed theoretical interpretation from the perspective of the EM framework is provided in Section 4.

### 3.1 NOISY-PAIR ROBUST SUPERVISED NON-CONTRASTIVE REPRESENTATION LEARNING

In PU learning, unreliable supervision poses a major obstacle to acquiring discriminative representations. To address this, *NcPU* integrates NoiSNCL with LDCE to facilitate the clustering effect in the representation space, thereby aligning intra-class representations.

**Preliminaries**. Compared with contrastive representation learning, non-contrastive representation learning acquire discriminative representations by only pulling same-class data closer to the anchor (intra-class representation alignment), thereby mitigating the impact of noisy different-class pairs. Without loss of generality, this paper adopts the classical non-contrastive representation learning framework BYOL (Grill et al., 2020). Given each sample $(\boldsymbol{x}, y)$, two different augmented views

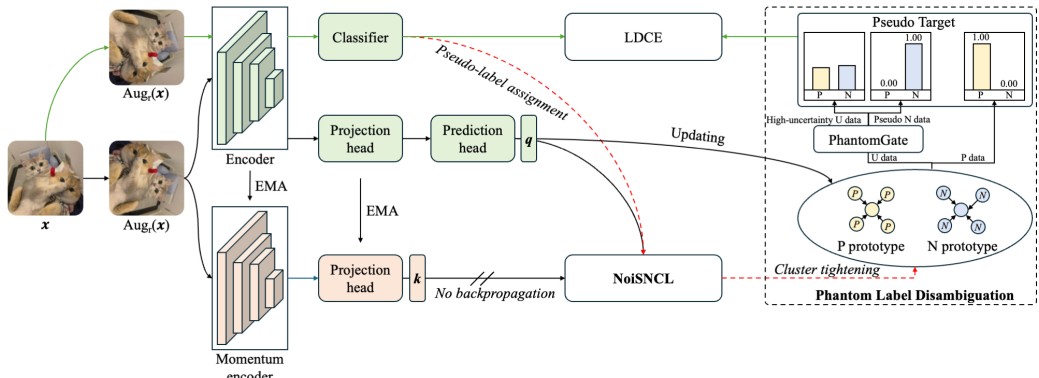

Figure 3: **The proposed *NcPU* framework**. NoiSNCL improves representations for label disambiguation, while PLD enhances supervision for representation learning.

$v = \text{Aug}_{\text{r}}(\boldsymbol{x})$ and $\boldsymbol{v}' = \text{Aug}_{\text{r}}(\boldsymbol{x})$ are produced by the image augmentation $\text{Aug}_{\text{r}}(\cdot)$. Because $\text{Aug}_{\text{r}}(\cdot)$ is stochastic, $\boldsymbol{v} \neq \boldsymbol{v}'$. From the first augmented view $\boldsymbol{v}$, the online network outputs the embedding $\boldsymbol{q}$. The target network outputs the other embedding $\boldsymbol{k}$ based on the augmented view $\boldsymbol{v}'$. Consistent with the original design of BYOL, the target network does not include the prediction head. The encoder and projection head of the target network are momentum-updated from the online network.

Self-supervised non-contrastive representation learning, such as BYOL, aims to pull the representations of the same sample from different views closer together:

$$\mathcal{L}_{\text{self-r}}(\boldsymbol{x}_i) = 2\big(1 - \langle \tilde{\boldsymbol{q}}_i, \tilde{\boldsymbol{k}}_i \rangle\big), \tag{4}$$

where $\tilde{\boldsymbol{q}} = \frac{\boldsymbol{q}}{\|\boldsymbol{q}\|_2}$ and $\tilde{\boldsymbol{k}} = \frac{\boldsymbol{k}}{\|\boldsymbol{k}\|_2}$ are the $L_2$-normalized embeddings. Similar to supervised contrastive loss (Khosla et al., 2020), the supervised non-contrastive loss ($\mathcal{L}_{\text{r}}$) can align intra-class representations through the incorporation of label information. Formally, $\mathcal{L}_{\text{r}}$ can be formalized as follows:

$$\mathcal{L}_{\text{r}}(\boldsymbol{x}_i, \boldsymbol{x}_j) = 2\big(1 - \langle \tilde{\boldsymbol{q}}_i, \tilde{\boldsymbol{k}}_j \rangle\big) \mathbb{1}\{y_i = y_j\}. \tag{5}$$

Minimizing Eq.(5) on $(\boldsymbol{x}_i, y_i)$ and $(\boldsymbol{x}_j, y_j)$ pulls representations of $\boldsymbol{x}_i$ and $\boldsymbol{x}_j$ closer if $y_i = y_j$.

**Gradient Analysis for Noisy Pairs**. In the PU learning setting, the ground-truth labels $y$ for unlabeled data are unavailable, thereby preventing the direct computation of Eq.(5). The use of pseudo labels $\tilde{y}$ generated by the classifier in *NcPU* inevitably introduces noisy pairs. These noisy pairs tend to dominate the representation learning process, as their gradient magnitudes overwhelm those from the clean pairs. For example, consider a clean pair $(\boldsymbol{x}_i, \boldsymbol{x}_j)$ with $y_i = y_j$ and a noisy pair $(\boldsymbol{x}_i, \boldsymbol{x}_m)$ with $y_i = \tilde{y}_m \neq y_m$, the representations may have $\tilde{\boldsymbol{q}}_i^\top \tilde{\boldsymbol{q}}_j \to 1$ and $\tilde{\boldsymbol{q}}_i^\top \tilde{\boldsymbol{q}}_m \approx 0$, which will wrongly pull together representations with incorrect labels and ultimately impair the learning of the clean pairs:

$$\left\| \frac{\partial \mathcal{L}_{\text{r}}(\boldsymbol{x}_i, \boldsymbol{x}_m)}{\partial \boldsymbol{q}_i} \right\|_2^2 = \frac{4}{\|\boldsymbol{q}_i\|_2^2}\big(1 - (\tilde{\boldsymbol{q}}_i^\top \tilde{\boldsymbol{q}}_m)^2\big) > \frac{4}{\|\boldsymbol{q}_i\|_2^2}\big(1 - (\tilde{\boldsymbol{q}}_i^\top \tilde{\boldsymbol{q}}_j)^2\big) = \left\| \frac{\partial \mathcal{L}_{\text{r}}(\boldsymbol{x}_i, \boldsymbol{x}_j)}{\partial \boldsymbol{q}_i} \right\|_2^2, \tag{6}$$

where the prediction head is regarded as an identity function for simplicity. The detailed proof is provided in Appendix B.

**Noisy-Pair Robust Supervised Non-Contrastive Loss**. The NoiSNCL, as shown in Eq.(7), is proposed to mitigate the problem of noisy pairs dominating the training process and is defined as follows:

$$\tilde{\mathcal{L}}_{\text{r}}(\boldsymbol{x}_i, \boldsymbol{x}_j) = 2\sqrt{1 - \langle \tilde{\boldsymbol{q}}_i, \tilde{\boldsymbol{k}}_j \rangle} \, \mathbb{1}\{y_i = y_j\}. \tag{7}$$

NoiSNCL still aims to align representations of same-class data, while the gradient magnitudes of clean pairs are greater than those of noisy pairs (Eq.(8)), the representation learning process is primarily driven by the clean pairs. Moreover, given that $\tilde{\mathcal{L}}_{\text{r}}(\boldsymbol{x}_i, \boldsymbol{x}_j) \geq \mathcal{L}_{\text{r}}(\boldsymbol{x}_i, \boldsymbol{x}_j)$, under some mild

assumptions, minimizing $\tilde{\mathcal{L}}_{\mathrm{r}}(\boldsymbol{x}_i, \boldsymbol{x}_j)$ in *NcPU* is equivalent to maximizing a lower bound of the likelihood function of the unlabeled data (Section 4).

$$\left\| \frac{\partial \tilde{\mathcal{L}}_{\mathrm{r}}(\boldsymbol{x}_i, \boldsymbol{x}_m)}{\partial \boldsymbol{q}_i} \right\|_2^2 = \frac{1}{\|\boldsymbol{q}_i\|_2^2} \left( \mathbf{1} + (\tilde{\boldsymbol{q}}_i^\top \tilde{\boldsymbol{q}}_m) \right) < \frac{1}{\|\boldsymbol{q}_i\|_2^2} \left( \mathbf{1} + (\tilde{\boldsymbol{q}}_i^\top \tilde{\boldsymbol{q}}_j) \right) = \left\| \frac{\partial \tilde{\mathcal{L}}_{\mathrm{r}}(\boldsymbol{x}_i, \boldsymbol{x}_j)}{\partial \boldsymbol{q}_i} \right\|_2^2. \quad (8)$$

Due to page limit, detailed proofs are provided in Appendix B.

## 3.2 PHANTOM LABEL DISAMBIGUATION FOR PU LEARNING

Based on the discriminative representations derived from noisy-pair robust supervised non-contrastive representation learning, the PLD is proposed to generate more accurate pseudo targets. This strategy is built upon class prototypes and enables regret-based label updating with the proposed PhantomGate. This strategy facilitates more conservative negative supervision while preserving the integrity of the prototype-based disambiguation process, where empirical evidence has demonstrated to be essential for effectively extending prototype-based label disambiguation to PU learning.

**Class-conditional Prototype Updating**. The class-conditional prototype embedding vector $\boldsymbol{\mu}_c$ serves as the representative embedding of class $c$:

$$\boldsymbol{\mu}_c = \text{Normalize}(\alpha \boldsymbol{\mu}_c + (1 - \alpha)\tilde{\boldsymbol{q}}), \quad (9)$$

where the prototype $\boldsymbol{\mu}_c$ is defined by the moving-average of the normalized embedding $\tilde{\boldsymbol{q}}$. The normalized embedding $\tilde{\boldsymbol{q}}$ corresponds to the representation of $\boldsymbol{x}$ produced by the online network, and the classifier assigns the label $c$ to the $\boldsymbol{x}$. Besides, $\alpha$ denotes a momentum hyperparameter.

**Phantom Pseudo Target Updating**. Benefiting from discriminative representations, the prototype can be employed to obtain more accurate pseudo targets:

$$\boldsymbol{s}' = \beta \boldsymbol{s}' + (1 - \beta)\boldsymbol{r}, \quad r_c = \begin{cases} 1 & \text{if} \quad c = \arg\max_j \tilde{\boldsymbol{q}}^\top \boldsymbol{\mu}_j, \\ 0 & \text{else,} \end{cases} \quad (10)$$

where $\beta$ denotes the momentum hyperparameter. However, this approach is effective when $\pi_p$ is explicitly provided as input (Yuan et al., 2025). In the absence of $\pi_p$, such a naive label disambiguation strategy in PU learning may lead to a trivial solution caused by the lack of negative information, namely $\boldsymbol{s}' = [1, 0]^\top$.

PhantomGate is proposed to prevent this trivial solution by injecting explicit negative supervision during training. Specifically, PhantomGate employs a threshold $\tau$ to assign $[0, 1]^\top$ to reliable pseudo negative samples, while simultaneously incorporating regret-based label updating. That is, when the model identifies that the pseudo target of a sample has been incorrectly set to $[0, 1]^\top$, PhantomGate allows the pseudo target to be updated starting from $\boldsymbol{s}'$ rather than resetting to $[0, 1]^\top$. The pseudo targets of positive data are fixed as $[1, 0]^\top$ throughout training. The pseudo targets $\boldsymbol{s}$ of unlabeled samples are initialized as $[0, 1]^\top$ and updated as follows:

$$\boldsymbol{s} = \begin{cases} [0, 1]^\top & f_1(\boldsymbol{x}) \geq \tau, \\ \boldsymbol{s}' & f_1(\boldsymbol{x}) < \tau, \end{cases} \quad (11)$$

where $f_c(\boldsymbol{x})$ denotes the $c$-th entry of the softmax output produced by the classifier.

To avoid manually setting $\tau$, the self-adaptive threshold (SAT) (Wang et al., 2023b) is introduced, which offers two key benefits: it starts low to supply clear supervision for more negative samples in early training, and gradually increases to filter out potentially incorrect negatives as training progresses. The global threshold $\tilde{\tau}$ and the local threshold ($\tilde{\boldsymbol{\rho}}$) are updated as follows:

$$\tilde{\tau} = \gamma \tilde{\tau} + (1 - \gamma) \frac{1}{b} \sum_{i=1}^b \max(f(\boldsymbol{x}_i)), \quad \tilde{\boldsymbol{\rho}}(c) = \gamma \tilde{\boldsymbol{\rho}}(c) + (1 - \gamma) \frac{1}{b} \sum_{i=1}^b f_c(\boldsymbol{x}_i), \quad (12)$$

where $b$ denotes the batch size, $\gamma$ is a momentum hyperparameter, and $\tilde{\boldsymbol{\rho}}(c)$ denotes the $c$-th entry of $\tilde{\boldsymbol{\rho}}$. Both $\tilde{\tau}$ and $\tilde{\boldsymbol{\rho}}(c)$ are initialized to 0.5. The $\tilde{\boldsymbol{\rho}}$ is used to modulate $\tilde{\tau}$ in a class-wise manner, and the final threshold $\tau$ can be obtained as follows:

$$\tau = \frac{\tilde{\boldsymbol{\rho}}(1)}{\max\{\tilde{\boldsymbol{\rho}}(0), \tilde{\boldsymbol{\rho}}(1)\}} \cdot \tilde{\tau}. \quad (13)$$

Finally, *NcPU* can be optimized as follows:

$$\mathcal{L} = \frac{1}{|\mathcal{P}|} \sum_{\boldsymbol{x}_i \in \mathcal{P}} \mathcal{L}_{\mathrm{c}} + \frac{1}{|\mathcal{U}|} \sum_{\boldsymbol{x}_i \in \mathcal{U}} \mathcal{L}_{\mathrm{c}} + w_{\mathrm{r}} \frac{1}{|\mathcal{D}|} \sum_{\boldsymbol{x}_i \in \mathcal{D}} \frac{1}{|\mathcal{Q}|} \sum_{\boldsymbol{x}_j \in \mathcal{Q}} \tilde{\mathcal{L}}_{\mathrm{r}}, \tag{14}$$

where $\mathcal{Q}$ denotes the set of same-class pairs associated with $\boldsymbol{x}_i$, and $w_{\mathrm{r}}$ is the weight hyperparameter. In practice, $\mathcal{L}$ is computed in a batch manner. An entropy regularization term is employed to stabilize training. The pseudocode of *NcPU* is provided in Appendix C.

## 4 THEORETICAL ANALYSIS OF *NcPU* BASED ON THE EM FRAMEWORK

By injecting the classifier predictions into the EM framework, the latent variables can be linked to the supervision from PU data, thereby enabling more accurate supervision for the unlabeled data during the iterative process. Different from previous works (Wang et al., 2024; Yuan et al., 2025), which conducted their theoretical analyses within the framework of supervised contrastive learning (Khosla et al., 2020) while neglecting the uniformity term, the theoretical analysis in this paper is instead built upon non-contrastive learning ($\tilde{\mathcal{L}}_{\mathrm{r}}$). By adopting the non-contrastive learning framework, this study avoids the complications arising from the uniformity term, thereby providing a more comprehensive theoretical perspective. Since the positive data have accurate labels, the following analysis focuses primarily on the unlabeled data. A more detailed derivation can be found in Appendix B.

**E-Step**. At the E-step, each unlabeled example is assigned to one specific cluster. Given a network $g(\cdot)$ parameterized by $\boldsymbol{\theta}$, the objective is to find $\boldsymbol{\theta}^*$ that maximizes the log-likelihood function:

$$\boldsymbol{\theta}^* = \arg\max_{\boldsymbol{\theta}} \sum_{\boldsymbol{x} \in \mathcal{U}} \log p(\boldsymbol{x}|\boldsymbol{\theta}) = \arg\max_{\boldsymbol{\theta}} \sum_{\boldsymbol{x} \in \mathcal{U}} \log \sum_{z \in \mathcal{Z}} p(\boldsymbol{x}, z|\boldsymbol{\theta}), \tag{15}$$

where $\mathcal{Z} = \{0, 1\}$ denotes the latent variable associated with the data. If the classifier predictions are injected into the EM framework: $p(z|\boldsymbol{x}, \boldsymbol{\theta}) = p(y|\boldsymbol{x}, \boldsymbol{\theta})$, and considering that the label of each sample is deterministic, we have $p(y|\boldsymbol{x}, \boldsymbol{\theta}) = \mathbb{1}(\tilde{y} = y)$. Then, $\boldsymbol{\theta}^*$ can be obtained as:

$$\boldsymbol{\theta}^* = \arg\max_{\boldsymbol{\theta}} \sum_{\boldsymbol{x} \in \mathcal{U}} \sum_{y \in \mathcal{Z}} \mathbb{1}(\tilde{y} = y) \log p(\boldsymbol{x}, y|\boldsymbol{\theta}). \tag{16}$$

**M-Step**. At the M-step, minimizing $\tilde{\mathcal{L}}_{\mathrm{r}}$ encourages embeddings to concentrate around their cluster centers (cluster tightening). For analytical convenience, all data are considered in each iteration:

$$\tilde{\mathcal{R}}_{\mathrm{r}}(\boldsymbol{x}) = \frac{1}{n_u} \sum_{\boldsymbol{x} \in \mathcal{U}} \frac{1}{|\mathcal{Q}|} \sum_{\boldsymbol{k}_+ \in \mathcal{Q}} \tilde{\mathcal{L}}_{\mathrm{r}} \geq \frac{1}{n_u} \sum_{\boldsymbol{x} \in \mathcal{U}} \frac{1}{|\mathcal{Q}|} \sum_{\boldsymbol{k}_+ \in \mathcal{Q}} \mathcal{L}_{\mathrm{r}} = \mathcal{R}_{\mathrm{r}}(\boldsymbol{x}). \tag{17}$$

Since the same-class peer set of $\boldsymbol{x}$ is constructed based on the classifier predictions, the unlabeled data can be divided into two subsets $\mathcal{S}_c \subseteq \mathcal{U}$ ($c \in \{0, 1\}$), where $\mathcal{S}_c = \{\boldsymbol{x}| \arg\max f(\boldsymbol{x}) = c, \boldsymbol{x} \in \mathcal{U}\}$. Then $\mathcal{R}_{\mathrm{r}}(\boldsymbol{x})$ can be reformulated as:

$$\mathcal{R}_{\mathrm{r}}(\boldsymbol{x}) \approx \frac{2}{n_u} \sum_{\mathcal{S}_c \in \mathcal{U}} \sum_{\boldsymbol{x} \in \mathcal{S}_c} \|\tilde{g}(\boldsymbol{x}) - \boldsymbol{\nu}_c\|^2, \tag{18}$$

where $\boldsymbol{\nu}_c$ denotes the mean center of $\mathcal{S}_c$. Since $n_u$ is usually large, we approximate $\frac{1}{|\mathcal{S}|} \approx \frac{1}{|\mathcal{Q}|}$. For simplicity, the augmentation operation is omitted and let $\tilde{q} = \tilde{g}(\boldsymbol{x})$. Under some mild assumptions, minimizing $\tilde{\mathcal{R}}_{\mathrm{r}}(\boldsymbol{x})$ is equivalent to maximizing a lower bound of the likelihood in Eq.(15).

**Theorem 1** *Assume the distribution of each class in the representation space follows a d-variate von Mises-Fisher (vMF) distribution, which leads to:* $h(\boldsymbol{x}|\tilde{\boldsymbol{\nu}}_c, \kappa) = c_d(\kappa)e^{\kappa \tilde{\boldsymbol{\nu}}_c^\top \tilde{g}(\boldsymbol{x})}$, *where* $\tilde{\boldsymbol{\nu}}_c = \boldsymbol{\nu}_c / \|\boldsymbol{\nu}_c\|$, $\kappa$ *is the concentration parameter, and* $c_d(\kappa)$ *is the normalization constant. Under the assumption of a uniform class prior, optimizing Eq.(18) and Eq.(15) is equivalent to maximizing $L_1$ and $L_2$ below, respectively.*

$$L_1 = \sum_{\mathcal{S}_c \in \mathcal{U}} \frac{|\mathcal{S}_c|}{n_u} \|\boldsymbol{\nu}_c\|^2 \leq \sum_{\mathcal{S}_c \in \mathcal{U}} \frac{|\mathcal{S}_c|}{n_u} \|\boldsymbol{\nu}_c\| = L_2. \tag{19}$$

A more detailed proof can be found in Appendix B. Theorem 1 indicates that minimizing $\tilde{\mathcal{R}}_{\mathrm{r}}(\boldsymbol{x})$ is equivalent to maximizing a lower bound of the likelihood in Eq.(15). The lower bound becomes tight when $\|\boldsymbol{\nu}_c\|$ is close to 1, which implies that the data with the same label are concentrated in the representation space.

Table 1: Results of different methods (mean±std). The best performance is highlighted in red.

| Method | Additional N Data | CIFAR-10 | | CIFAR-100 | | STL-10 | | ABCD | | xBD | |
|---|---|---|---|---|---|---|---|---|---|---|---|
| | | OA | F1 | OA | F1 | OA | F1 | OA | F1 | OA | F1 |
| CE | | $60.45^{\pm0.1}$ | $2.42^{\pm0.4}$ | $50.36^{\pm0.0}$ | $1.86^{\pm0.2}$ | $50.30^{\pm0.0}$ | $1.19^{\pm0.2}$ | $55.70^{\pm0.2}$ | $20.93^{\pm0.4}$ | $84.08^{\pm0.2}$ | $25.70^{\pm2.4}$ |
| uPU | | $65.52^{\pm0.2}$ | $26.82^{\pm0.9}$ | $61.44^{\pm0.9}$ | $43.12^{\pm2.1}$ | $57.08^{\pm0.4}$ | $25.88^{\pm1.3}$ | $83.76^{\pm2.1}$ | $81.47^{\pm2.9}$ | $86.82^{\pm0.3}$ | $55.43^{\pm2.8}$ |
| nnPU | | $87.29^{\pm0.5}$ | $83.71^{\pm0.6}$ | $72.00^{\pm0.8}$ | $74.93^{\pm0.4}$ | $80.62^{\pm0.1}$ | $79.28^{\pm0.2}$ | $87.73^{\pm0.4}$ | $88.36^{\pm0.3}$ | $82.60^{\pm0.6}$ | $59.66^{\pm0.7}$ |
| vPU | | $85.94^{\pm0.6}$ | $82.98^{\pm0.9}$ | $69.01^{\pm1.2}$ | $70.78^{\pm0.2}$ | $75.76^{\pm5.5}$ | $70.52^{\pm10.4}$ | $84.06^{\pm3.0}$ | $84.13^{\pm3.4}$ | $73.60^{\pm1.8}$ | $50.30^{\pm1.2}$ |
| ImbPU | | $87.29^{\pm0.4}$ | $83.80^{\pm0.4}$ | $72.07^{\pm0.7}$ | $75.05^{\pm0.6}$ | $80.68^{\pm0.6}$ | $79.41^{\pm0.6}$ | $88.14^{\pm0.6}$ | $88.69^{\pm0.5}$ | $82.51^{\pm0.5}$ | $59.72^{\pm0.5}$ |
| TEDn | | $86.29^{\pm2.4}$ | $80.70^{\pm4.6}$ | $69.85^{\pm0.9}$ | $61.73^{\pm1.9}$ | $66.26^{\pm4.9}$ | $49.90^{\pm10.7}$ | $88.90^{\pm0.9}$ | $89.10^{\pm0.9}$ | $85.40^{\pm0.8}$ | $52.65^{\pm4.6}$ |
| PUET | | $78.51^{\pm0.4}$ | $73.85^{\pm0.5}$ | $62.81^{\pm0.2}$ | $71.09^{\pm0.1}$ | $75.36^{\pm0.2}$ | $73.56^{\pm0.1}$ | $78.09^{\pm2.9}$ | $66.52^{\pm24.9}$ | $74.92^{\pm0.1}$ | $38.38^{\pm0.6}$ |
| HolisticPU | | $84.20^{\pm2.1}$ | $78.10^{\pm3.9}$ | $64.01^{\pm6.5}$ | $51.94^{\pm15.1}$ | $72.81^{\pm6.4}$ | $66.06^{\pm14.9}$ | $65.49^{\pm1.5}$ | $51.60^{\pm1.5}$ | $81.98^{\pm4.1}$ | $53.35^{\pm2.4}$ |
| DistPU | | $85.29^{\pm2.6}$ | $83.96^{\pm1.2}$ | $67.63^{\pm0.8}$ | $73.68^{\pm0.8}$ | $85.62^{\pm1.5}$ | $85.41^{\pm0.9}$ | $86.25^{\pm1.7}$ | $87.36^{\pm1.2}$ | $82.94^{\pm0.8}$ | $57.58^{\pm0.2}$ |
| PiCO | | $89.72^{\pm0.0}$ | $87.40^{\pm0.0}$ | $69.98^{\pm0.4}$ | $72.71^{\pm0.3}$ | $60.71^{\pm0.6}$ | $71.04^{\pm0.3}$ | $74.07^{\pm2.2}$ | $79.27^{\pm1.3}$ | $49.36^{\pm0.5}$ | $39.52^{\pm0.2}$ |
| LaGAM | ✓ | $95.78^{\pm0.5}$ | $94.90^{\pm0.6}$ | $84.82^{\pm0.1}$ | $84.42^{\pm0.2}$ | $88.64^{\pm0.0}$ | $88.50^{\pm0.1}$ | $75.90^{\pm0.4}$ | $75.38^{\pm0.6}$ | $79.14^{\pm1.5}$ | $58.78^{\pm1.7}$ |
| WSC | | $90.55^{\pm0.3}$ | $87.92^{\pm0.8}$ | $75.39^{\pm2.1}$ | $73.76^{\pm4.0}$ | $79.06^{\pm4.5}$ | $74.16^{\pm7.0}$ | $80.10^{\pm2.8}$ | $76.12^{\pm4.3}$ | $84.89^{\pm0.8}$ | $62.17^{\pm1.3}$ |
| NcPU(ours) | | $97.36^{\pm0.1}$ | $96.67^{\pm0.2}$ | $88.28^{\pm0.6}$ | $88.14^{\pm0.9}$ | $91.40^{\pm0.4}$ | $90.82^{\pm0.6}$ | $91.10^{\pm0.6}$ | $91.21^{\pm0.5}$ | $87.60^{\pm1.0}$ | $64.84^{\pm1.0}$ |
| Supervised | ✓ | $96.96^{\pm0.2}$ | $96.24^{\pm0.2}$ | $89.65^{\pm0.3}$ | $89.78^{\pm0.4}$ | — | — | $92.00^{\pm0.2}$ | $91.96^{\pm0.2}$ | $88.47^{\pm0.3}$ | $73.32^{\pm0.4}$ |

## 5 EXPERIMENTS

### 5.1 EXPERIMENTAL SETTINGS

**Datasets**. To evaluate the performance of the proposed *NcPU*, five datasets are employed: three prevalent benchmark datasets (CIFAR-10 (Krizhevsky et al., 2009), CIFAR-100 (Krizhevsky et al., 2009) and STL-10 (Coates et al., 2011)), as well as two remote sensing post-disaster building damage mapping datasets (ABCD (Fujita et al., 2017) and xBD (Gupta et al., 2019)), which can be used to identify damaged buildings after disasters. Experiments on ABCD and xBD demonstrate the significant potential of PU learning in the field of HADR. More details regarding ABCD and xBD are presented in Appendix D. Each dataset was partitioned into two non-overlapping subsets, denoted as positive and negative, according to their category labels. For the training process, 1000 positive samples were utilized from CIFAR-10, CIFAR-100, and STL-10, 300 from ABCD, and 500 from xBD, respectively. A comprehensive summary of the dataset statistics is presented in Appendix D.

**Baselines**. To evaluate the effectiveness of *NcPU*, we compare it against eleven baseline methods: uPU (Du Plessis et al., 2015), nnPU (Kiryo et al., 2017), vPU (Chen et al., 2020), ImbPU (Su et al., 2021), TEDn (Garg et al., 2021), PUET (Wilton et al., 2022), HolisticPU (Wang et al., 2023a), DistPU (Jiang et al., 2023), PiCO (Wang et al., 2024), LaGAM (Long et al., 2024), and WSC (Zhou et al., 2025). In the CE method, unlabeled data are treated as negative samples. Representation learning modules are also incorporated into PiCO, LaGAM, and WSC. Nevertheless, LaGAM requires auxiliary negative samples, whereas WSC relies on additional pre-estimated parameters. PiCO, originally designed for partial label learning, exhibits suboptimal performance when applied to PU learning tasks. For *NcPU*, all momentum hyperparameters are set to 0.99, and the $w_r$ is set to 50 for all datasets. More detailed implementation settings are provided in Appendix F. ResNet-18 (He et al., 2016) is adopted as the backbone for all methods. For methods that require $\pi_p$, $\pi_p$ is estimated using KM2 (Ramaswamy et al., 2016).

**Metrics**. Overall accuracy (OA) and F1 score are adopted as the primary metrics. Precision, recall, and area under receiver operating characteristic curve are provided in Appendix G. All experiments are repeated three times, and both the mean and standard deviation are reported.

### 5.2 MAIN RESULTS

***NcPU* achieves the best performance.** The results of all methods are presented in Table 1, where the proposed *NcPU* achieves the best performance across all datasets. Notably, *NcPU* does not require additional negative samples or pre-estimated parameters. Compared with other methods that also do not rely on additional negative samples, *NcPU* achieves improvements in OA of 6.81%, 12.89%, 5.78%, 2.20%, and 0.78% on CIFAR-10, CIFAR-100, STL-10, ABCD, and xBD, respectively. Although LaGAM achieves the second-best performance on three benchmark datasets, it requires additional negative samples as input. As an algorithm originally designed for partial label

Table 2: Ablation and comparative analyses on CIFAR-100 dataset. NCL: Non-contrastive Loss. LD: Label Disambiguation.

| NCL | LD | OA | F1 | P | R |
|---|---|---|---|---|---|
|  | $s$ | $61.54^{\pm7.8}$ | $40.58^{\pm22.9}$ | $84.36^{\pm7.1}$ | $30.69^{\pm25.0}$ |
| $\tilde{\mathcal{L}}_r$ |  | $50.27^{\pm0.1}$ | $1.09^{\pm0.4}$ | $97.97^{\pm1.8}$ | $0.55^{\pm0.2}$ |
| $\mathcal{L}_{self-r}$ | $s$ | $73.22^{\pm1.7}$ | $72.75^{\pm1.6}$ | $74.10^{\pm2.2}$ | $71.47^{\pm1.9}$ |
| $\mathcal{L}_r$ | $s$ | $84.58^{\pm0.8}$ | $85.90^{\pm0.6}$ | $79.12^{\pm1.1}$ | $93.96^{\pm0.3}$ |
| $\tilde{\mathcal{L}}_r$ | $s'$ | $75.14^{\pm2.7}$ | $79.91^{\pm1.7}$ | $67.15^{\pm2.6}$ | $98.73^{\pm0.5}$ |
| $\tilde{\mathcal{L}}_r$ | $s'$+SAT | $50.25^{\pm0.0}$ | $1.01^{\pm0.1}$ | $97.85^{\pm3.7}$ | $0.51^{\pm0.1}$ |
| $\tilde{\mathcal{L}}_r$ | $s$ | $88.28^{\pm0.6}$ | $88.14^{\pm0.9}$ | $89.12^{\pm1.7}$ | $87.27^{\pm3.2}$ |

Table 3: Performance Analysis of $\tilde{\mathcal{L}}_r$ under risk estimation with the real $\pi_p$ and supervised methods.

| Method | CIFAR-10 | | CIFAR-100 | |
|---|---|---|---|---|
|  | OA | F1 | OA | F1 |
| uPU | $69.43^{\pm0.3}$ | $41.32^{\pm1.0}$ | $61.68^{\pm0.9}$ | $44.18^{\pm2.6}$ |
| uPU+$\tilde{\mathcal{L}}_r$ | $97.35^{\pm0.1}$ | $96.66^{\pm0.2}$ | $83.71^{\pm1.6}$ | $81.40^{\pm2.2}$ |
| nnPU | $83.25^{\pm0.2}$ | $76.94^{\pm0.5}$ | $71.22^{\pm0.5}$ | $68.12^{\pm1.0}$ |
| nnPU+$\tilde{\mathcal{L}}_r$ | $97.03^{\pm0.2}$ | $96.37^{\pm0.2}$ | $87.81^{\pm0.3}$ | $87.23^{\pm0.4}$ |
| Supervised+$\mathcal{L}_r$ | $98.53^{\pm0.0}$ | $98.17^{\pm0.1}$ | $94.45^{\pm0.1}$ | $94.52^{\pm0.1}$ |
| Supervised+$\tilde{\mathcal{L}}_r$ | $98.75^{\pm0.0}$ | $98.43^{\pm0.1}$ | $94.56^{\pm0.1}$ | $94.64^{\pm0.1}$ |

learning, PiCO exhibits inferior performance compared with *NcPU*. Finally, it is observed that *NcPU* achieves results comparable to, or even surpass, those of its supervised counterpart, demonstrating the effectiveness of noisy-pair robust representation alignment in PU learning.

**NcPU learns more discriminative representations.** The t-SNE visualization of representations on the training data demonstrates the learning ability of the PU learning methods under unreliable supervision (Figure 2). As shown in Figure 2, compared with other PU learning methods, *NcPU* produces more discriminative representations through noisy-pair robust intra-class representation alignment.

**NcPU has broad application potential in HADR.** *NcPU* achieved the best performance on both the ABCD and xBD datasets. Among them, the xBD dataset, which encompasses nineteen disaster events with global coverage, highlights the remarkable application potential of *NcPU*.

## 5.3 ABLATION STUDIES AND ANALYSES

$\tilde{\mathcal{L}}_r$ **and** $s$ **can benefit each other**. Due to the influence of non-discriminative features, $s$ alone is insufficient, as shown in Table 2, whereas its integration with $\tilde{\mathcal{L}}_r$ leads to notable improvements. Furthermore, this mutually beneficial effect can be theoretically justified from the perspective of the EM algorithm, as detailed in Section 4.

$\tilde{\mathcal{L}}_r$ **demonstrates robustness against noisy pairs**. The results in Table 2 corroborate the theoretical analysis, demonstrating the robustness of $\tilde{\mathcal{L}}_r$ to noisy pairs, with $\tilde{\mathcal{L}}_r + s$ achieving 88.28% OA compared to 84.58% for $\mathcal{L}_r + s$. $\tilde{\mathcal{L}}_r$ is robust to noisy pairs without losing performance in supervised learning tasks, achieving results similar to $\mathcal{L}_r$ (Supervised+$\mathcal{L}_r$ *vs.*Supervised+$\tilde{\mathcal{L}}_r$ in Table 3).

**PhantomGate plays an important role in label disambiguation**. As shown in Table 2, in the absence of unambiguous negative supervision, class-conditional prototype label disambiguation ($s'$) tends to yield a trivial solution, resulting in high recall but low precision for the positive class. When SAT is employed to introduce negative supervision, the inaccurate negative supervision improves the precision of the positive class but substantially reduces its recall. In contrast, PhantomGate enables the model to discard inaccurately selected negative samples, thereby achieving a better balance between precision and recall of the positive class.

$\tilde{\mathcal{L}}_r$ **significantly enhances PU learning methods**. Risk estimation methods, which have been validated both theoretically and empirically as the most simply and robust approaches, are utilized as baselines to reduce the impact of hyperparameters and heuristic techniques. As shown in Table 3, compared with uPU and nnPU, the methods employing $\tilde{\mathcal{L}}_r$ achieve superior results, highlighting the importance of learning discriminative representations. The t-SNE visualization of learned representations are shown in Figure 4. While nnPU serves as the foundation of

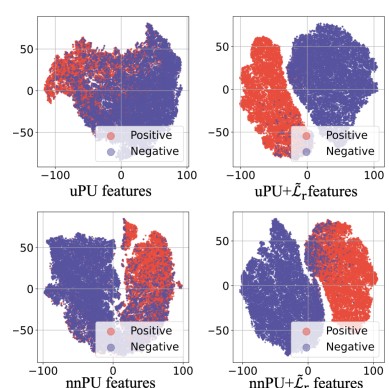

Figure 4: t-SNE visualizations of the representations learned by risk estimation methods on CIFAR-10 training dataset.

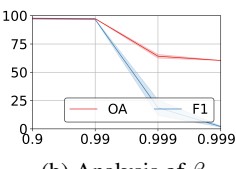 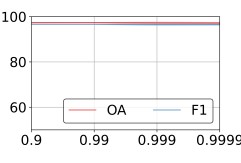 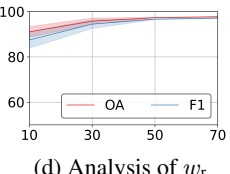

(a) Analysis of $\alpha$     (b) Analysis of $\beta$     (c) Analysis of $\gamma$     (d) Analysis of $w_{\mathrm{r}}$

Figure 5: Analyses of hyperparameters on the CIFAR-10 dataset.

deep PU learning by constraining the negative risk to be non-negative, we show that uPU+$\tilde{\mathcal{L}}_{\mathrm{r}}$ can achieve superior performance without this constraint. **Remarkably, $\tilde{\mathcal{L}}_{\mathrm{r}}$ enables simple PU methods to achieve highly competitive performance**, this insight opens a promising direction for deep PU learning. More experiments are presented in Appendix H.

**Analysis of hyperparameters**. As shown in Figure 5, *NcPU* is insensitive to $\alpha$ and $\gamma$. A smaller $\beta$ accelerates the update of pseudo targets, whereas a larger $w_{\mathrm{r}}$ facilitates the learning of more discriminative representations, thereby enhancing label disambiguation. Additional experiments are provided in Appendix H. Overall, *NcPU* demonstrates robustness to hyperparameters, as the same parameters are used across all five datasets: $\alpha = \beta = \gamma = 0.99$ and $w_{\mathrm{r}} = 50$.

**Analysis of training stability**. While the gradients of $\tilde{\mathcal{L}}_{\mathrm{r}}$ do not vanish and include the term $\frac{1}{\sqrt{1-\tilde{q}_i^\top \tilde{k}_j}}$, they do not lead to significant overfitting or training instability: (1) Benefiting from the asymmetric architecture and random data augmentation, $\tilde{q}_i \neq \tilde{k}_j$ is ensured, which can enable the numerical stability of the gradient; (2) As $\tilde{q}_i^\top \tilde{k}_j$ approaches 1, the gradient magnitude becomes $\frac{2}{\|q_i\|_2^2}$ instead of infinity, within a finite number of training iterations, $\tilde{\mathcal{L}}_{\mathrm{r}}$ does not lead to notable overfitting or training instability; (3) The training stability can also be empirically verified (Figure 6): Taking the results on CIFAR-10 dataset as an example, *NcPU* has achieved promising performance at the 400th epoch. After extended training for a longer period, the results of *NcPU* does not exhibit overfitting or instability; (4) The numerical stability can be improved by constraining the value of $\tilde{q}_i^\top \tilde{k}_j$ (e.g., $[10^{-4}, 1 - 10^{-4}]$).

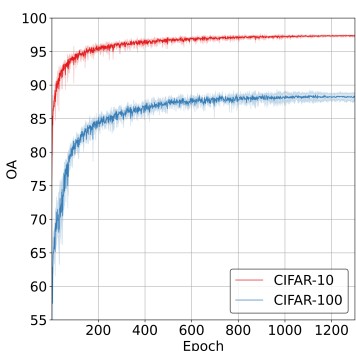

Figure 6: **OA curve of *NcPU* during the training process.** *NcPU* exhibits stable training performance over a prolonged training process.

## 6 RELATED WORKS

**Positive-Unlabeled Learning**. A straightforward approach is to identify reliable negatives from the unlabeled set and then train a supervised classifier on positives and these negatives (Gong et al., 2018; Yu et al., 2004), but its performance heavily depends on the correctness of the selected samples. Recent studies try to directly estimate supervision from all PU data, such as cost-sensitive based methods (Li et al., 2021), label disambiguation based methods (Zhang et al., 2019), risk estimation based methods (Kiryo et al., 2017; Zhao et al., 2022; Su et al., 2021; Wilton et al., 2022; Zhao et al., 2023b), density ratio estimation-based methods (Kato et al., 2019), and variational principle methods (Chen et al., 2020; Zhao et al., 2023a). Recent studies (Long et al., 2024; Yuan et al., 2025) leverage a contrastive learning module to obtain better representations, but still suffer from noisy pairs and require either auxiliary negatives (Long et al., 2024) or pre-estimated $\pi_p$ (Yuan et al., 2025). In contrast, the proposed *NcPU* is robust to noisy pairs and free from such requirements.

**Contrastive and Non-contrastive Representation Learning**. Self-supervised learning has recently made significant progress in acquiring high-quality representations. Depending on the use of negative pairs, self-supervised learning methods are broadly categorized into contrastive and non-contrastive learning: contrastive losses encourage embeddings of similar images to be aligned (alignment) while pushing apart those of different images (uniformity) (He et al., 2020; Wang & Isola, 2020; Huang et al., 2023a; Lu et al., 2023); non-contrastive losses only align embeddings of similar images (Grill et al., 2020; Chen & He, 2021; Guo et al., 2025). Beyond self-supervised set-

tings, SupCon was proposed to extend contrastive learning by incorporating full supervision (Khosla et al., 2020). Recent works actively explore the application of contrastive learning to weakly supervised tasks (Li et al., 2022; Huang et al., 2023b; Zhang et al., 2024; Wang et al., 2022; 2024; Yuan et al., 2025). However, most of these methods rely on supervised contrastive learning with pseudo labels, which suffer from the problem of noisy pairs. The CoTAP Loss (Wen et al., 2025) is proposed to align the representations of semantically similar objects in self-supervised dense representation learning, and the negative effect of noises is alleviated by assigning higher weights to sample pairs with top scores. The WSC framework (Zhou et al., 2025), a graph-theoretic weakly supervised contrastive learning method, introduces continuous semantic similarity; however, it requires pre-estimated parameters as input. In contrast, *NcPU*, which builds on non-contrastive learning, is robust to noisy pairs and does not rely on such requirements.

## 7   CONCLUSION

This paper identifies the challenge of learning discriminative representations as the key bottleneck in PU learning. Theoretical analysis demonstrates that noisy pairs dominate representation learning optimization, while the proposed NoiSNCL and PLD form an EM-inspired framework that iteratively benefits each other. Extensive evaluations on both benchmark datasets and remote sensing building damage mapping tasks validate the effectiveness of the proposed *NcPU*. Applications of binary classification with only positive data annotated can benefit from this study, and we expect future research to extend this framework beyond image classification. Masked image modeling will be incorporated to harness more powerful backbone frameworks. Beyond PU learning, the framework holds promise for broader applications in more weakly supervised learning scenarios.

## ETHICS STATEMENT

This study complies with the ICLR Code of Ethics. The study does not involve human subjects, and no personally identifiable information was collected or processed. All datasets used in this work are publicly available and widely adopted in the research community. We have carefully considered potential risks of harm that may arise from our methodology or findings, and we believe the insights primarily benefit scientific and educational purposes. While our model may be applied in various domains, we encourage responsible usage and highlight that it should not be deployed in sensitive contexts without thorough evaluation of fairness, bias, privacy, and security concerns. We confirm that this work adheres to research integrity standards, including accurate reporting of methods and results, avoidance of conflicts of interest, and compliance with relevant legal and ethical guidelines.

## REPRODUCIBILITY STATEMENT

A code repository (`https://github.com/Hengwei-Zhao96/NcPU`) is provided to facilitate the reproduction of the experimental results. Theoretical results, including assumptions and complete proofs, are presented in detail in the main text and the appendix. All datasets used in this work are publicly available and have been widely adopted in the research community.

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

## A    MORE T-SNE VISUALIZATIONS OF LEARNED REPRESENTATIONS

CE denotes the training strategy that employs cross-entropy loss, where unlabeled data are assumed to be negative samples. Compared to supervised features, the positive and negative features from other methods (nnPU (Kiryo et al., 2017), DistPU (Jiang et al., 2023), HolisticPU (Wang et al., 2023a), TEDn (Garg et al., 2021), WSC (Zhou et al., 2025)) exhibit significant overlap on the training dataset (Figure 7).

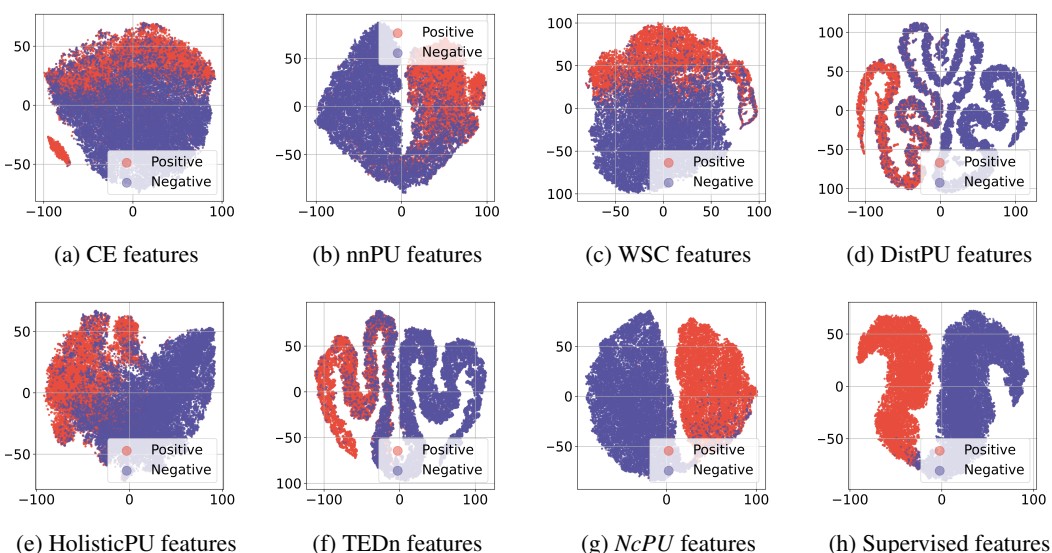

Figure 7: **t-SNE visualizations of the representations learned by different PU learning methods on CIFAR-10 training dataset**. Compared with supervised features, the positive and negative features obtained from other methods (nnPU, DistPU, HolisticPU, TEDn, WSC) exhibit substantial overlap, indicating that the primary bottleneck in PU learning lies in acquiring discriminative representations. In contrast, the representations produced by *NcPU* demonstrate discriminative ability.

## B    THEORETICAL ANALYSIS

### B.1    DERIVATION OF GRADIENTS AND GRADIENT MAGNITUDES

The supervised non-contrastive loss ($\mathcal{L}_{\mathrm{r}}$) and noisy-pair robust supervised non-contrastive loss ($\tilde{\mathcal{L}}_{\mathrm{r}}$) are defined as follows:

$$\mathcal{L}_{\mathrm{r}}(\boldsymbol{x}_i, \boldsymbol{x}_j) = 2\big(1 - 1\langle \tilde{\boldsymbol{q}}_i, \tilde{\boldsymbol{k}}_j \rangle\big)\mathbb{1}\{y_i = y_j\},$$

$$\tilde{\mathcal{L}}_{\mathrm{r}}(\boldsymbol{x}_i, \boldsymbol{x}_j) = 2\sqrt{1 - \langle \tilde{\boldsymbol{q}}_i, \tilde{\boldsymbol{k}}_j \rangle}\,\mathbb{1}\{y_i = y_j\},$$

where $\tilde{\boldsymbol{q}} = \frac{\boldsymbol{q}}{\|\boldsymbol{q}\|_2}$ and $\tilde{\boldsymbol{k}} = \frac{\boldsymbol{k}}{\|\boldsymbol{k}\|_2}$. Only the case $\mathbb{1}\{y_i = y_j\} = 1$ is considered because the loss is not calculated when $\mathbb{1}\{y_i = y_j\} = 0$. The prediction head is regarded as an identity function for simplicity.

The gradients of $\mathcal{L}_{\mathrm{r}}(\boldsymbol{x}_i, \boldsymbol{x}_j)$ can be obtained as follows:

$$
\begin{aligned}
\frac{\partial \mathcal{L}_{\mathrm{r}}(\boldsymbol{x}_i, \boldsymbol{x}_j)}{\partial \boldsymbol{q}_i} &= -2 \frac{\partial \tilde{\boldsymbol{q}}_i^\top \tilde{\boldsymbol{k}}_j}{\partial \boldsymbol{q}_i} \\
&= -2 \frac{\partial \tilde{\boldsymbol{q}}_i}{\partial \boldsymbol{q}_i} \tilde{\boldsymbol{k}}_j \\
&= -2 \Big( \frac{1}{\sqrt{\boldsymbol{q}_i^\top \boldsymbol{q}_i}} \frac{\partial \boldsymbol{q}_i}{\partial \boldsymbol{q}_i} - \frac{1}{\boldsymbol{q}_i^\top \boldsymbol{q}_i} \frac{\partial \sqrt{\boldsymbol{q}_i^\top \boldsymbol{q}_i}}{\partial \boldsymbol{q}_i} \boldsymbol{q}_i^\top \Big) \tilde{\boldsymbol{k}}_j \\
&= -2 \Big( \frac{1}{\sqrt{\boldsymbol{q}_i^\top \boldsymbol{q}_i}} \mathbf{I} - \frac{1}{\sqrt{\boldsymbol{q}_i^\top \boldsymbol{q}_i}} \tilde{\boldsymbol{q}}_i \tilde{\boldsymbol{q}}_i^\top \Big) \tilde{\boldsymbol{k}}_j \\
&= -\frac{2}{\|\boldsymbol{q}_i\|_2} \big( \mathbf{I} - \tilde{\boldsymbol{q}}_i \tilde{\boldsymbol{q}}_i^\top \big) \tilde{\boldsymbol{k}}_j.
\end{aligned}
\tag{20}
$$

The gradients of $\tilde{\mathcal{L}}_{\mathrm{r}}(\boldsymbol{x}_i, \boldsymbol{x}_j)$ can be obtained as follows:

$$
\begin{aligned}
\frac{\partial \tilde{\mathcal{L}}_{\mathrm{r}}(\boldsymbol{x}_i, \boldsymbol{x}_j)}{\partial \boldsymbol{q}_i} &= \frac{-1}{\sqrt{1 - \tilde{\boldsymbol{q}}_i^\top \tilde{\boldsymbol{k}}_j}} \frac{\partial \tilde{\boldsymbol{q}}_i^\top \tilde{\boldsymbol{k}}_j}{\partial \boldsymbol{q}_i} \\
&= \frac{-1}{\|\boldsymbol{q}_i\|_2 \sqrt{1 - \tilde{\boldsymbol{q}}_i^\top \tilde{\boldsymbol{k}}_j}} \big( \mathbf{I} - \tilde{\boldsymbol{q}}_i \tilde{\boldsymbol{q}}_i^\top \big) \tilde{\boldsymbol{k}}_j.
\end{aligned}
\tag{21}
$$

Considering that the following equation holds:

$$
\begin{aligned}
\big( \mathbf{I} - \tilde{\boldsymbol{q}}_i \tilde{\boldsymbol{q}}_i^\top \big)^\top &= \mathbf{I} - \big( \tilde{\boldsymbol{q}}_i \tilde{\boldsymbol{q}}_i^\top \big)^\top \\
&= \mathbf{I} - (\tilde{\boldsymbol{q}}_i^\top)^\top \tilde{\boldsymbol{q}}_i^\top \\
&= \mathbf{I} - \tilde{\boldsymbol{q}}_i \tilde{\boldsymbol{q}}_i^\top.
\end{aligned}
\tag{22}
$$

$$
\begin{aligned}
\big( \mathbf{I} - \tilde{\boldsymbol{q}}_i \tilde{\boldsymbol{q}}_i^\top \big)^\top \big( \mathbf{I} - \tilde{\boldsymbol{q}}_i \tilde{\boldsymbol{q}}_i^\top \big) &= \big( \mathbf{I} - \tilde{\boldsymbol{q}}_i \tilde{\boldsymbol{q}}_i^\top \big) \big( \mathbf{I} - \tilde{\boldsymbol{q}}_i \tilde{\boldsymbol{q}}_i^\top \big) \\
&= \mathbf{I} - \tilde{\boldsymbol{q}}_i \tilde{\boldsymbol{q}}_i^\top - \tilde{\boldsymbol{q}}_i \tilde{\boldsymbol{q}}_i^\top + \tilde{\boldsymbol{q}}_i \tilde{\boldsymbol{q}}_i^\top \\
&= \mathbf{I} - \tilde{\boldsymbol{q}}_i \tilde{\boldsymbol{q}}_i^\top.
\end{aligned}
\tag{23}
$$

The gradient magnitudes of $\mathcal{L}_{\mathrm{r}}(\boldsymbol{x}_i, \boldsymbol{x}_j)$ can be obtained as follows:

$$
\begin{aligned}
\Big\| \frac{\partial \mathcal{L}_{\mathrm{r}}(\boldsymbol{x}_i, \boldsymbol{x}_j)}{\partial \boldsymbol{q}_i} \Big\|_2^2 &= \Big( \frac{\partial \mathcal{L}_{\mathrm{r}}(\boldsymbol{x}_i, \boldsymbol{x}_j)}{\partial \boldsymbol{q}_i} \Big)^\top \Big( \frac{\partial \mathcal{L}_{\mathrm{r}}(\boldsymbol{x}_i, \boldsymbol{x}_j)}{\partial \boldsymbol{q}_i} \Big) \\
&= \frac{4}{\|\boldsymbol{q}_i\|_2^2} \tilde{\boldsymbol{k}}_j^\top \big( \mathbf{I} - \tilde{\boldsymbol{q}}_i \tilde{\boldsymbol{q}}_i^\top \big)^\top \big( \mathbf{I} - \tilde{\boldsymbol{q}}_i \tilde{\boldsymbol{q}}_i^\top \big) \tilde{\boldsymbol{k}}_j \\
&= \frac{4}{\|\boldsymbol{q}_i\|_2^2} \tilde{\boldsymbol{k}}_j^\top \big( \mathbf{I} - \tilde{\boldsymbol{q}}_i \tilde{\boldsymbol{q}}_i^\top \big) \tilde{\boldsymbol{k}}_j \\
&= \frac{4}{\|\boldsymbol{q}_i\|_2^2} \big( \tilde{\boldsymbol{k}}_j^\top \tilde{\boldsymbol{k}}_j - \tilde{\boldsymbol{k}}_j^\top \tilde{\boldsymbol{q}}_i \tilde{\boldsymbol{q}}_i^\top \tilde{\boldsymbol{k}}_j \big) \\
&= \frac{4}{\|\boldsymbol{q}_i\|_2^2} \big( 1 - (\tilde{\boldsymbol{q}}_i^\top \tilde{\boldsymbol{k}}_j)^2 \big) \\
&= \frac{4}{\|\boldsymbol{q}_i\|_2^2} \big( 1 - (\tilde{\boldsymbol{q}}_i^\top \tilde{\boldsymbol{q}}_j)^2 \big).
\end{aligned}
\tag{24}
$$

The magnitudes of gradient of $\tilde{\mathcal{L}}_{\mathrm{r}}(\boldsymbol{x}_i, \boldsymbol{x}_j)$ can be obtained as follows:

$$
\begin{aligned}
\left\| \frac{\partial \tilde{\mathcal{L}}_{\mathrm{r}}(\boldsymbol{x}_i, \boldsymbol{x}_j)}{\partial \boldsymbol{q}_i} \right\|_2^2 &= \Big( \frac{\partial \tilde{\mathcal{L}}_{\mathrm{r}}(\boldsymbol{x}_i, \boldsymbol{x}_j)}{\partial \boldsymbol{q}_i} \Big)^\top \Big( \frac{\partial \tilde{\mathcal{L}}_{\mathrm{r}}(\boldsymbol{x}_i, \boldsymbol{x}_j)}{\partial \boldsymbol{q}_i} \Big) \\
&= \frac{1}{\|\boldsymbol{q}_i\|_2^2 (1 - \tilde{\boldsymbol{q}}_i^\top \tilde{\boldsymbol{k}}_j)} \tilde{\boldsymbol{k}}_j^\top (\mathbf{I} - \tilde{\boldsymbol{q}}_i \tilde{\boldsymbol{q}}_i^\top)^\top (\mathbf{I} - \tilde{\boldsymbol{q}}_i \tilde{\boldsymbol{q}}_i^\top) \tilde{\boldsymbol{k}}_j \\
&= \frac{1}{\|\boldsymbol{q}_i\|_2^2 (1 - \tilde{\boldsymbol{q}}_i^\top \tilde{\boldsymbol{k}}_j)} \big( 1 - (\tilde{\boldsymbol{q}}_i^\top \tilde{\boldsymbol{k}}_j)^2 \big) \\
&= \frac{1}{\|\boldsymbol{q}_i\|_2^2 (1 - \tilde{\boldsymbol{q}}_i^\top \tilde{\boldsymbol{k}}_j)} \big( 1 - (\tilde{\boldsymbol{q}}_i^\top \tilde{\boldsymbol{k}}_j) \big) \big( 1 + (\tilde{\boldsymbol{q}}_i^\top \tilde{\boldsymbol{k}}_j) \big) \\
&= \frac{1}{\|\boldsymbol{q}_i\|_2^2} \big( 1 + (\tilde{\boldsymbol{q}}_i^\top \tilde{\boldsymbol{k}}_j) \big) \\
&= \frac{1}{\|\boldsymbol{q}_i\|_2^2} \big( 1 + (\tilde{\boldsymbol{q}}_i^\top \tilde{\boldsymbol{q}}_j) \big).
\end{aligned}
\tag{25}
$$

### B.2 DETAILED DERIVATION OF *NcPU* IN EM FRAMEWORK

At the E-step, each unlabeled example is assigned to one specific cluster. At the M-step, minimizing $\tilde{\mathcal{R}}_{\mathrm{r}}(\boldsymbol{x})$ encourages the embeddings to concentrate around their respective cluster centers. The detailed theoretical interpretation is provided below.

**E-Step**. Given a network $g(\cdot)$ parameterized by $\boldsymbol{\theta}$, the objective is to find $\boldsymbol{\theta}^*$ that maximizes the log-likelihood function:

$$
\boldsymbol{\theta}^* = \arg\max_{\boldsymbol{\theta}} \sum_{\boldsymbol{x} \in \mathcal{U}} \log p(\boldsymbol{x}|\boldsymbol{\theta}) = \arg\max_{\boldsymbol{\theta}} \sum_{\boldsymbol{x} \in \mathcal{U}} \log \sum_{z \in \mathcal{Z}} p(\boldsymbol{x}, z|\boldsymbol{\theta}),
$$

where $\mathcal{Z} = \{0, 1\}$ denotes the latent variable associated with data. Since this function is difficult to optimize directly, a surrogate function is used to lower-bound it:

$$
\begin{aligned}
\sum_{\boldsymbol{x} \in \mathcal{U}} \log \sum_{z \in \mathcal{Z}} p(\boldsymbol{x}, z|\boldsymbol{\theta}) &= \sum_{\boldsymbol{x} \in \mathcal{U}} \log \sum_{z \in \mathcal{Z}} Q(z) \frac{p(\boldsymbol{x}, z|\boldsymbol{\theta})}{Q(z)} \\
&\geq \sum_{\boldsymbol{x} \in \mathcal{U}} \sum_{z \in \mathcal{Z}} Q(z) \log \frac{p(\boldsymbol{x}, z|\boldsymbol{\theta})}{Q(z)},
\end{aligned}
\tag{26}
$$

where $Q(z)$ denotes a distribution over $z$'s ($\sum_{z \in \mathcal{Z}} Q(z) = 1$). To ensure that the equality holds, $\frac{p(\boldsymbol{x}, z|\boldsymbol{\theta})}{Q(z)}$ must be a constant. Then, we have:

$$
Q(z) = \frac{p(\boldsymbol{x}, z|\boldsymbol{\theta})}{\sum_{z \in \mathcal{Z}} p(\boldsymbol{x}, z|\boldsymbol{\theta})} = \frac{p(\boldsymbol{x}, z|\boldsymbol{\theta})}{p(\boldsymbol{x}|\boldsymbol{\theta})} = p(z|\boldsymbol{x}, \boldsymbol{\theta}),
\tag{27}
$$

which corresponds to the posterior class probability: $p(z|\boldsymbol{x}, \boldsymbol{\theta}) = p(y|\boldsymbol{x}, \boldsymbol{\theta})$. At this step, the classifier predictions are injected into the EM framework, linking the latent variables with the supervision from PU data. Considered the predicted label $\tilde{y} = \arg\max f(\boldsymbol{x})$, and considering that the label of each sample is deterministic, we have $p(y|\boldsymbol{x}, \boldsymbol{\theta}) = \mathbb{1}(\tilde{y} = y)$. Then, $\boldsymbol{\theta}^*$ can be obtained as:

$$
\boldsymbol{\theta}^* = \arg\max_{\boldsymbol{\theta}} \sum_{\boldsymbol{x} \in \mathcal{U}} \sum_{y \in \mathcal{Z}} \mathbb{1}(\tilde{y} = y) \log p(\boldsymbol{x}, y|\boldsymbol{\theta}).
\tag{28}
$$

**M-Step**. At the M-step, the goal is to maximize the likelihood presented in Eq.(28). We will demonstrate that under some mild assumptions, minimizing $\tilde{\mathcal{R}}_{\mathrm{r}}(\boldsymbol{x})$ is equivalent to maximizing a lower bound of Eq.(28). For analytical convenience, all unlabeled data are considered in each iteration:

$$
\mathcal{R}_{\mathrm{r}}(\boldsymbol{x}) = \frac{1}{n_u} \sum_{\boldsymbol{x} \in \mathcal{U}} \left\{ \frac{2}{|\mathcal{Q}|} \sum_{\boldsymbol{k}_+ \in \mathcal{Q}} \Big( 1 - \tilde{\boldsymbol{q}}^\top \tilde{\boldsymbol{k}}_+ \Big) \right\},
\tag{29}
$$

$$\tilde{\mathcal{R}}_{\mathrm{r}}(\boldsymbol{x}) = \frac{1}{n_u} \sum_{\boldsymbol{x} \in \mathcal{U}} \left\{ \frac{2}{|\mathcal{Q}|} \sum_{\boldsymbol{k}_+ \in \mathcal{Q}} \sqrt{1 - \tilde{\boldsymbol{q}}^{\top} \tilde{\boldsymbol{k}}_+} \right\}, \tag{30}$$

where $\mathcal{Q}$ corresponds to the positive peer set of the example $\boldsymbol{x}$. Since the positive peer set of $\boldsymbol{x}$ is constructed based on the supervised model predictions $\tilde{y} = \arg\max f(\boldsymbol{x})$, the unlabeled data can be divided into two subsets $\mathcal{S}_c \subseteq \mathcal{U}$ ($c \in \{0, 1\}$), where $\mathcal{S}_c = \{\boldsymbol{x} | \arg\max f(\boldsymbol{x}) = c, \boldsymbol{x} \in \mathcal{U}\}$. Then the $\mathcal{R}_{\mathrm{r}}(\boldsymbol{x})$ can be reformulated as follows:

$$
\begin{aligned}
\mathcal{R}_{\mathrm{r}}(\boldsymbol{x}) &= \frac{2}{n_u} \sum_{\boldsymbol{x} \in \mathcal{U}} \frac{1}{2|\mathcal{Q}|} \sum_{\boldsymbol{k}_+ \in \mathcal{Q}} \left\| \tilde{\boldsymbol{q}} - \tilde{\boldsymbol{k}}_+ \right\|^2 \\
&\approx \frac{2}{n_u} \sum_{\mathcal{S}_c \in \mathcal{U}} \frac{1}{2|\mathcal{S}_c|} \sum_{\boldsymbol{x},\boldsymbol{x}' \in \mathcal{S}_c} \left\| \tilde{g}(\boldsymbol{x}) - \tilde{g}(\boldsymbol{x}') \right\|^2 \\
&= \frac{2}{n_u} \sum_{\mathcal{S}_c \in \mathcal{U}} \frac{1}{2|\mathcal{S}_c|} \sum_{\boldsymbol{x} \in \mathcal{S}_c} \sum_{\boldsymbol{x}' \in \mathcal{S}_c} \left( \|\tilde{g}(\boldsymbol{x})\|^2 - 2\tilde{g}(\boldsymbol{x})^{\top} \tilde{g}(\boldsymbol{x}') + \|\tilde{g}(\boldsymbol{x}')\|^2 \right) \\
&= \frac{2}{n_u} \sum_{\mathcal{S}_c \in \mathcal{U}} \frac{1}{|\mathcal{S}_c|} \sum_{\boldsymbol{x} \in \mathcal{S}_c} \left( |\mathcal{S}_c| - \tilde{g}(\boldsymbol{x})^{\top} \left( \sum_{\boldsymbol{x}' \in \mathcal{S}_c} \tilde{g}(\boldsymbol{x}') \right) \right) \\
&= \frac{2}{n_u} \sum_{\mathcal{S}_c \in \mathcal{U}} \frac{1}{|\mathcal{S}_c|} \sum_{\boldsymbol{x} \in \mathcal{S}_c} \left( |\mathcal{S}_c| - \tilde{g}(\boldsymbol{x})^{\top} (|\mathcal{S}_c|\boldsymbol{\nu}_c) \right) \\
&= \frac{2}{n_u} \sum_{\mathcal{S}_c \in \mathcal{U}} \left( |\mathcal{S}_c| - \left( \sum_{\boldsymbol{x} \in \mathcal{S}_c} \tilde{g}(\boldsymbol{x}) \right)^{\top} \boldsymbol{\nu}_c \right) \\
&= \frac{2}{n_u} \sum_{\mathcal{S}_c \in \mathcal{U}} |\mathcal{S}_c| \left( 1 - \|\boldsymbol{\nu}_c\|^2 \right) \\
&= \frac{2}{n_u} \sum_{\mathcal{S}_c \in \mathcal{U}} \left( |\mathcal{S}_c| - 2 \left( \sum_{\boldsymbol{x} \in \mathcal{S}_c} \tilde{g}(\boldsymbol{x}) \right)^{\top} \boldsymbol{\nu}_c + |\mathcal{S}_c| \|\boldsymbol{\nu}_c\|^2 \right) \\
&= \frac{2}{n_u} \sum_{\mathcal{S}_c \in \mathcal{U}} \sum_{\boldsymbol{x} \in \mathcal{S}_c} \left( \|\tilde{g}(\boldsymbol{x})\|^2 - 2\tilde{g}(\boldsymbol{x})^{\top} \boldsymbol{\nu}_c + \|\boldsymbol{\nu}_c\|^2 \right) \\
&= \frac{2}{n_u} \sum_{\mathcal{S}_c \in \mathcal{U}} \sum_{\boldsymbol{x} \in \mathcal{S}_c} \|\tilde{g}(\boldsymbol{x}) - \boldsymbol{\nu}_c\|^2,
\end{aligned} \tag{31}
$$

where $\boldsymbol{\nu}_c$ represents the mean center of $\mathcal{S}_c$. For simplicity, the augmentation operation is omitted and let $\tilde{q} = \tilde{g}(\boldsymbol{x})$. Since $n_u$ is usually large, we approximate $\frac{1}{|\mathcal{S}_c|} \approx \frac{1}{|\mathcal{Q}|}$.

Assume the distribution of each class in the representation space is a $d$-variate von Mises-Fisher (vMF) distribution, which leads to: $h(\boldsymbol{x}|\tilde{\boldsymbol{\nu}}_c, \kappa) = c_d(\kappa) e^{\kappa \tilde{\boldsymbol{\nu}}_c^{\top} \tilde{g}(\boldsymbol{x})}$, where $\tilde{\boldsymbol{\nu}}_c = \boldsymbol{\nu}_c / \|\boldsymbol{\nu}_c\|$, $\kappa$ is the concentration parameter, and $c_d(\kappa)$ is the normalization constant. Under the assumption of a uniform class prior, optimizing Eq.(31) and Eq.(28) is equivalent to maximizing $L_1$ and $L_2$ below, respectively.

$$L_1 = \sum_{\mathcal{S}_c \in \mathcal{U}} \frac{|\mathcal{S}_c|}{n_u} \|\boldsymbol{\nu}_c\|^2 \leq \sum_{\mathcal{S}_c \in \mathcal{U}} \frac{|\mathcal{S}_c|}{n_u} \|\boldsymbol{\nu}_c\| = L_2. \tag{32}$$

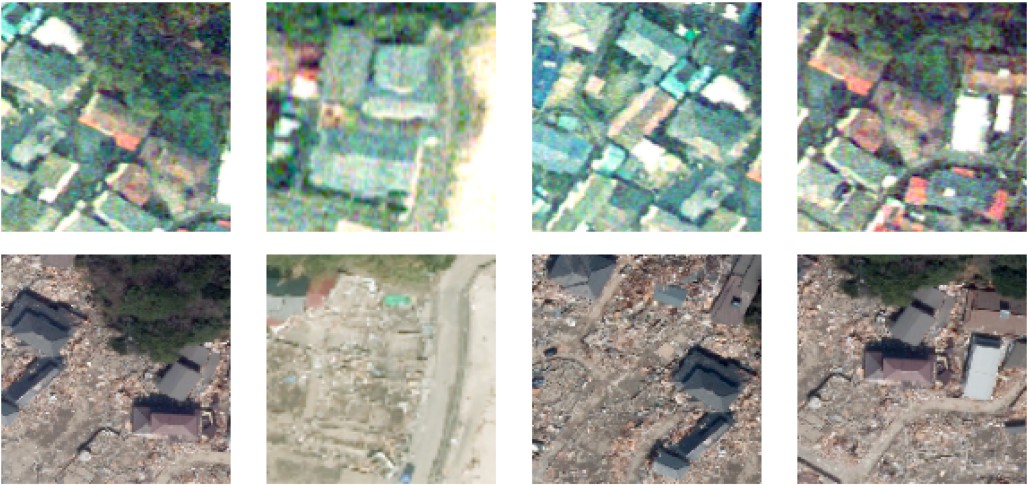

Figure 8: Examples from the ABCD dataset: the first row shows pre-disaster imagery, and the second row shows post-disaster imagery.

The lower bound becomes tight when $\|\boldsymbol{\nu}_c\|$ is close to 1, which implies that the data with the same label are concentrated in the representation space. More detailed proofs are as follows:

$$
\begin{aligned}
\arg\min_{\boldsymbol{\theta}} \mathcal{R}_{\mathrm{r}}(\boldsymbol{x}) &= \arg\min_{\boldsymbol{\theta}} \frac{2}{n_u} \sum_{\mathcal{S}_c \in \mathcal{U}} \sum_{\boldsymbol{x} \in \mathcal{S}_c} \|\tilde{g}(\boldsymbol{x}) - \boldsymbol{\nu}_c\|^2 \\
&= \arg\min_{\boldsymbol{\theta}} \frac{2}{n_u} \sum_{\mathcal{S}_c \in \mathcal{U}} \sum_{\boldsymbol{x} \in \mathcal{S}_c} \left( \|\tilde{g}(\boldsymbol{x})\|^2 - 2\tilde{g}(\boldsymbol{x})^\top \boldsymbol{\nu}_c + \|\boldsymbol{\nu}_c\|^2 \right) \\
&= \arg\min_{\boldsymbol{\theta}} \frac{4}{n_u} \sum_{\mathcal{S}_c \in \mathcal{U}} \left( |\mathcal{S}_c| - |\mathcal{S}_c| \, \|\boldsymbol{\nu}_c\|^2 \right) \\
&= \arg\max_{\boldsymbol{\theta}} \sum_{\mathcal{S}_c \in \mathcal{U}} \frac{|\mathcal{S}_c|}{n_u} \|\boldsymbol{\nu}_c\|^2 .
\end{aligned}
\tag{33}
$$

$$
\begin{aligned}
\arg\max_{\boldsymbol{\theta}} \sum_{\boldsymbol{x} \in \mathcal{U}} \sum_{y \in \mathcal{Z}} \mathbb{1}(\tilde{y} = y) \log p(\boldsymbol{x}, y|\boldsymbol{\theta}) &= \arg\max_{\boldsymbol{\theta}} \sum_{\boldsymbol{x} \in \mathcal{U}} \sum_{y \in \mathcal{Z}} \mathbb{1}(\tilde{y} = y) \log p(\boldsymbol{x}|y, \boldsymbol{\theta}) \\
&= \arg\max_{\boldsymbol{\theta}} \sum_{\mathcal{S}_c \in \mathcal{U}} \sum_{\boldsymbol{x} \in \mathcal{S}_c} \log p(\boldsymbol{x}|y = c, \boldsymbol{\theta}) \\
&= \arg\max_{\boldsymbol{\theta}} \sum_{\mathcal{S}_c \in \mathcal{U}} \sum_{\boldsymbol{x} \in \mathcal{S}_c} \kappa \tilde{\boldsymbol{\nu}}_c^\top \tilde{g}(\boldsymbol{x}) \\
&= \arg\max_{\boldsymbol{\theta}} \sum_{\mathcal{S}_c \in \mathcal{U}} \frac{|\mathcal{S}_c|}{n_u} \|\boldsymbol{\nu}_c\| .
\end{aligned}
\tag{34}
$$

Given that

$$
\mathcal{L}_{\mathrm{r}}(\boldsymbol{x}_i, \boldsymbol{x}_j) \leq \tilde{\mathcal{L}}_{\mathrm{r}}(\boldsymbol{x}_i, \boldsymbol{x}_j),
\tag{35}
$$

we can obtain

$$
\mathcal{R}_{\mathrm{r}}(\boldsymbol{x}) \leq \tilde{\mathcal{R}}_{\mathrm{r}}(\boldsymbol{x}).
\tag{36}
$$

In other words, minimizing $\tilde{\mathcal{R}}_{\mathrm{r}}(\boldsymbol{x})$ is also equivalent to maximizing a lower bound of Eq.(28). Given that *NcPU* can be interpreted from the perspective of the EM framework, its non-contrastive learning module and classification module can mutually enhance each other during the iterative process, allowing *NcPU* to converge to a (local) optimum.

---

**Algorithm 1:** Pseudocode of *NcPU* (one epoch).

1 **Input:** Training dataset $\mathcal{D}$, classifier $f$, online network $g$, target network $g'$, initialized pseudo targets $\boldsymbol{s}_i$, initialized class prototypes $\boldsymbol{\mu}_c$, momentum hyperparameters $\alpha, \beta, \gamma$.

2 **for** $iter = 1, 2, \ldots,$ **do**

3     sample a mini-batch $B$ from $\mathcal{D}$
    // classifier prediction

4     $\tilde{Y} = \{\tilde{y}_i = \arg\max_c f_c(\boldsymbol{x}_i) | \boldsymbol{x}_i \in B\}$
    // online embeddings generation

5     $B_q = \{\boldsymbol{q}_i = g(\mathrm{Aug}_\mathrm{r}(\boldsymbol{x}_i)) | \boldsymbol{x}_i \in B\}$
    // class-conditional prototype updating

6     **for** $\boldsymbol{x}_i \in B$ **do**

7         $\boldsymbol{\mu}_c = \mathrm{Normalize}(\alpha\boldsymbol{\mu}_c + (1-\alpha)\tilde{\boldsymbol{q}}_i)$, if $\tilde{y}_i = c$

8     **end**
    // update the target network

9     momentum update $g'$ by using $g$
    // target embeddings generation

10     $B_k = \{\boldsymbol{k}_i = g'(\mathrm{Aug}_\mathrm{r}(\boldsymbol{x}_i)) | \boldsymbol{x}_i \in B\}$
    // self-adaptive threshold updating

11     $\tau = \frac{\tilde{\boldsymbol{\rho}}(1)}{\max\{\tilde{\boldsymbol{\rho}}(0), \tilde{\boldsymbol{\rho}}(1)\}} \cdot \tilde{\tau}$
    // phantom pseudo target updating

12     **for** $\boldsymbol{x}_i \in B$ **do**

13         $\boldsymbol{r}_i = \mathrm{OneHot}(\arg\max_j \tilde{\boldsymbol{q}}_i^\top \boldsymbol{\mu}_j)$

14         $\boldsymbol{s}_i' = \beta\boldsymbol{s}_i' + (1-\beta)\boldsymbol{r}_i$

15         $\boldsymbol{s}_i = \begin{cases} [0,1]^\top & f_1(\boldsymbol{x}_i) \geq \tau \\ \boldsymbol{s}' & f_1(\boldsymbol{x}_i) < \tau \end{cases}$

16     **end**
    // network updating

17     minimize loss $\mathcal{L}$

18 **end**

---

## C   Pseudocode of *NcPU*

The pseudocode of *NcPU* is summarized in Algorithm 1.

## D   Detailed Description of Datasets

To demonstrate the promising applicability of the proposed *NcPU* in the field of HADR, two remote sensing building damage mapping datasets are utilized: ABCD (Fujita et al., 2017) and xBD (Gupta et al., 2019). The ABCD dataset is a single-hazard dataset annotated to determine whether buildings were washed away by a tsunami. It comprises both pre-disaster and post-disaster imagery, as illustrated in Figure 8. The spatial resolution of the ABCD dataset is 40 cm. The xBD dataset is a large-scale, multi-hazard dataset encompassing building damages caused by six different disaster types worldwide (earthquake, wildfire, volcano, storm, flooding, and tsunami). It contains both pre-disaster and post-disaster imagery; however, to ensure distinction from ABCD, only the post-disaster imagery from xBD is employed in this study. Specifically, in processing xBD, each building is buffered by a distance equal to 0.5 times its size, and the minimum bounding rectangle of each buffer is cropped and used as training data. For classification, the categories "Destroyed" and "Major Damage" are regarded as the positive class. More detailed dataset statistics are presented in Table 4.

## E   Implementation Details of PiCO and WSC

For PiCO Wang et al. (2022; 2024), this paper treats unlabeled data as data associated with a coarse candidate label set. During training, only the pseudo labels of the unlabeled data are updated, while positive samples are consistently trained with their ground-truth labels. At the beginning of training,

Table 4: Summary of datasets.

| Dataset | $n_p$ | $n_u$ | # Testing Data | $\pi_p$ | Positive Class | Input Size |
|---|---|---|---|---|---|---|
| CIFAR-10 | 1000 | 40000 | 10000 | 0.4 | 0,1,8,9 | $3 \times 32 \times 32$ |
| CIFAR-100 | 1000 | 40000 | 10000 | 0.5 | Animal | $3 \times 32 \times 32$ |
| STL-10 | 1000 | 90000 | 8000 | - | 0,2,3,8,9 | $3 \times 96 \times 96$ |
| ABCD | 300 | 4000 | 3377 | 0.5 | washed-away | $6 \times 128 \times 128$ |
| xBD | 500 | 20000 | 37682 | 0.4 | 2,3 | $3 \times 64 \times 64$ |

Table 5: Results on CIFAR-10 and CIFAR-100 datasets (mean±std).

| Method | Additional N Data | CIFAR-10 | | | | | CIFAR-100 | | | | |
|---|---|---|---|---|---|---|---|---|---|---|---|
| | | OA | F1 | P | R | AUC | OA | F1 | P | R | AUC |
| CE | | $60.45^{\pm0.1}$ | $2.42^{\pm0.4}$ | $92.62^{\pm6.5}$ | $1.22^{\pm0.2}$ | $59.16^{\pm1.9}$ | $50.36^{\pm0.0}$ | $1.86^{\pm0.2}$ | $80.92^{\pm4.4}$ | $0.94^{\pm0.1}$ | $62.36^{\pm0.4}$ |
| uPU | | $65.52^{\pm0.2}$ | $26.82^{\pm0.9}$ | $88.73^{\pm0.7}$ | $15.80^{\pm0.7}$ | $50.43^{\pm0.9}$ | $61.44^{\pm0.9}$ | $43.12^{\pm2.1}$ | $82.09^{\pm1.0}$ | $29.25^{\pm1.8}$ | $71.19^{\pm0.7}$ |
| nnPU | | $87.29^{\pm0.5}$ | $83.71^{\pm0.6}$ | $85.89^{\pm1.2}$ | $81.65^{\pm1.0}$ | $89.77^{\pm0.3}$ | $72.00^{\pm0.8}$ | $74.93^{\pm0.4}$ | $67.85^{\pm1.1}$ | $83.69^{\pm0.9}$ | $78.61^{\pm0.2}$ |
| vPU | | $85.94^{\pm0.6}$ | $82.98^{\pm0.9}$ | $80.42^{\pm0.9}$ | $85.74^{\pm2.2}$ | $92.95^{\pm0.8}$ | $69.01^{\pm1.2}$ | $70.78^{\pm0.2}$ | $67.10^{\pm2.6}$ | $75.07^{\pm3.6}$ | $75.51^{\pm1.1}$ |
| ImbPU | | $87.29^{\pm0.4}$ | $83.80^{\pm0.4}$ | $85.53^{\pm1.0}$ | $82.13^{\pm0.1}$ | $89.76^{\pm0.1}$ | $72.07^{\pm0.7}$ | $75.05^{\pm0.6}$ | $67.82^{\pm0.9}$ | $84.02^{\pm1.2}$ | $78.30^{\pm0.6}$ |
| TEDn | | $86.29^{\pm2.4}$ | $80.70^{\pm4.6}$ | $91.63^{\pm1.7}$ | $72.47^{\pm8.2}$ | $94.53^{\pm0.2}$ | $69.85^{\pm0.9}$ | $61.73^{\pm1.9}$ | $84.48^{\pm1.4}$ | $48.67^{\pm2.5}$ | $81.83^{\pm0.6}$ |
| PUET | | $78.51^{\pm0.4}$ | $73.85^{\pm0.5}$ | $71.94^{\pm0.4}$ | $75.86^{\pm0.6}$ | - | $62.81^{\pm0.2}$ | $71.09^{\pm0.1}$ | $58.14^{\pm0.1}$ | $91.45^{\pm0.2}$ | - |
| HolisticPU | | $84.20^{\pm2.1}$ | $78.10^{\pm3.9}$ | $87.31^{\pm2.4}$ | $70.96^{\pm7.4}$ | $93.08^{\pm1.1}$ | $64.01^{\pm6.5}$ | $51.94^{\pm15.1}$ | $75.71^{\pm3.0}$ | $41.40^{\pm18.4}$ | $73.89^{\pm4.6}$ |
| DistPU | | $85.29^{\pm2.6}$ | $83.96^{\pm2.2}$ | $74.77^{\pm4.1}$ | $95.86^{\pm0.9}$ | $95.80^{\pm0.6}$ | $67.63^{\pm0.8}$ | $73.68^{\pm0.8}$ | $62.07^{\pm0.5}$ | $90.64^{\pm1.6}$ | $77.53^{\pm1.8}$ |
| PiCO | | $89.72^{\pm0.1}$ | $87.40^{\pm0.0}$ | $85.74^{\pm0.4}$ | $89.13^{\pm0.4}$ | $95.61^{\pm0.1}$ | $69.98^{\pm0.4}$ | $72.71^{\pm0.3}$ | $66.66^{\pm0.4}$ | $79.97^{\pm0.4}$ | $76.47^{\pm0.3}$ |
| LaGAM | ✓ | $95.78^{\pm0.5}$ | $94.90^{\pm0.6}$ | $92.03^{\pm1.5}$ | $97.96^{\pm0.5}$ | $99.11^{\pm0.1}$ | $84.82^{\pm0.1}$ | $84.42^{\pm0.2}$ | $86.73^{\pm0.6}$ | $82.23^{\pm0.8}$ | $92.33^{\pm0.3}$ |
| WSC | | $90.55^{\pm0.3}$ | $87.92^{\pm0.8}$ | $89.99^{\pm2.5}$ | $86.08^{\pm3.6}$ | $96.23^{\pm0.2}$ | $75.39^{\pm2.1}$ | $73.76^{\pm4.0}$ | $78.79^{\pm2.1}$ | $69.76^{\pm8.1}$ | $83.35^{\pm1.8}$ |
| NcPU(ours) | | $97.36^{\pm0.1}$ | $96.67^{\pm0.2}$ | $97.53^{\pm0.4}$ | $95.82^{\pm0.6}$ | $99.30^{\pm0.1}$ | $88.28^{\pm0.6}$ | $88.14^{\pm0.9}$ | $89.12^{\pm1.7}$ | $87.27^{\pm3.2}$ | $94.99^{\pm0.1}$ |
| Supervised | ✓ | $96.96^{\pm0.2}$ | $96.24^{\pm0.2}$ | $95.43^{\pm0.6}$ | $97.06^{\pm0.2}$ | $99.58^{\pm0.0}$ | $89.65^{\pm0.3}$ | $89.78^{\pm0.4}$ | $88.63^{\pm0.5}$ | $90.97^{\pm1.3}$ | $95.99^{\pm0.2}$ |

Table 6: Results on STL-10 and ABCD datasets (mean±std).

| Method | Additional N Data | STL-10 | | | | | ABCD | | | | |
|---|---|---|---|---|---|---|---|---|---|---|---|
| | | OA | F1 | P | R | AUC | OA | F1 | P | R | AUC |
| CE | | $50.30^{\pm0.0}$ | $1.19^{\pm0.2}$ | $98.48^{\pm2.6}$ | $0.60^{\pm0.1}$ | $75.38^{\pm0.5}$ | $55.70^{\pm0.2}$ | $20.93^{\pm0.4}$ | $95.65^{\pm0.8}$ | $11.75^{\pm0.3}$ | $87.18^{\pm0.5}$ |
| uPU | | $57.08^{\pm0.4}$ | $25.88^{\pm1.3}$ | $94.67^{\pm0.5}$ | $14.99^{\pm0.9}$ | $72.84^{\pm1.7}$ | $83.76^{\pm2.1}$ | $81.47^{\pm2.9}$ | $94.27^{\pm0.4}$ | $71.83^{\pm4.6}$ | $92.67^{\pm0.6}$ |
| nnPU | | $80.62^{\pm0.1}$ | $79.28^{\pm0.2}$ | $85.14^{\pm0.2}$ | $74.19^{\pm0.5}$ | $87.96^{\pm0.2}$ | $87.73^{\pm0.4}$ | $88.36^{\pm0.3}$ | $83.93^{\pm1.0}$ | $93.29^{\pm0.9}$ | $94.08^{\pm0.0}$ |
| vPU | | $75.76^{\pm5.5}$ | $70.52^{\pm10.4}$ | $87.93^{\pm2.5}$ | $60.13^{\pm14.7}$ | $84.85^{\pm1.1}$ | $84.06^{\pm3.0}$ | $84.13^{\pm3.4}$ | $84.46^{\pm9.1}$ | $85.50^{\pm12.0}$ | $93.19^{\pm1.0}$ |
| ImbPU | | $80.68^{\pm0.6}$ | $79.41^{\pm0.6}$ | $85.00^{\pm0.7}$ | $74.52^{\pm0.6}$ | $87.89^{\pm0.2}$ | $88.14^{\pm0.6}$ | $88.69^{\pm0.5}$ | $84.56^{\pm0.8}$ | $93.25^{\pm0.4}$ | $94.33^{\pm0.3}$ |
| TEDn | | $66.26^{\pm4.9}$ | $49.90^{\pm10.7}$ | $95.00^{\pm1.5}$ | $34.35^{\pm10.2}$ | $86.77^{\pm1.8}$ | $88.90^{\pm0.9}$ | $89.10^{\pm0.9}$ | $87.35^{\pm1.6}$ | $90.96^{\pm1.9}$ | $94.95^{\pm0.8}$ |
| PUET | | $75.36^{\pm0.2}$ | $73.56^{\pm0.1}$ | $79.35^{\pm0.6}$ | $68.56^{\pm0.3}$ | - | $78.09^{\pm2.9}$ | $66.52^{\pm24.9}$ | $61.97^{\pm25.0}$ | $71.94^{\pm24.3}$ | - |
| HolisticPU | | $72.81^{\pm6.4}$ | $66.06^{\pm14.9}$ | $85.95^{\pm10.6}$ | $58.26^{\pm25.5}$ | $86.74^{\pm1.4}$ | $65.49^{\pm1.5}$ | $51.60^{\pm1.5}$ | $87.69^{\pm11.6}$ | $36.97^{\pm3.9}$ | $82.87^{\pm10.3}$ |
| DistPU | | $85.62^{\pm1.5}$ | $85.41^{\pm0.9}$ | $87.13^{\pm5.0}$ | $84.00^{\pm3.2}$ | $92.19^{\pm0.2}$ | $86.25^{\pm1.7}$ | $87.36^{\pm1.2}$ | $80.84^{\pm2.8}$ | $95.11^{\pm1.3}$ | $95.30^{\pm0.3}$ |
| PiCO | | $60.71^{\pm0.6}$ | $71.04^{\pm0.3}$ | $56.26^{\pm0.4}$ | $96.36^{\pm0.3}$ | $78.80^{\pm1.0}$ | $74.07^{\pm2.2}$ | $79.27^{\pm1.3}$ | $66.00^{\pm2.0}$ | $99.25^{\pm0.2}$ | $83.65^{\pm1.5}$ |
| LaGAM | ✓ | $88.64^{\pm0.0}$ | $88.50^{\pm0.1}$ | $89.60^{\pm0.5}$ | $87.43^{\pm0.6}$ | $95.25^{\pm0.1}$ | $75.90^{\pm0.4}$ | $75.38^{\pm0.6}$ | $76.85^{\pm0.2}$ | $73.99^{\pm1.0}$ | $83.87^{\pm1.4}$ |
| WSC | | $79.06^{\pm4.5}$ | $74.16^{\pm7.0}$ | $95.40^{\pm0.7}$ | $61.11^{\pm10.0}$ | $92.89^{\pm0.7}$ | $80.10^{\pm2.8}$ | $76.12^{\pm4.3}$ | $94.33^{\pm0.6}$ | $63.98^{\pm6.2}$ | $94.07^{\pm0.4}$ |
| NcPU(ours) | | $91.40^{\pm0.4}$ | $90.82^{\pm0.6}$ | $97.38^{\pm0.8}$ | $85.11^{\pm1.6}$ | $96.52^{\pm0.1}$ | $91.10^{\pm0.6}$ | $91.21^{\pm0.5}$ | $89.89^{\pm1.2}$ | $92.58^{\pm0.5}$ | $96.01^{\pm0.3}$ |
| Supervised | ✓ | - | - | - | - | - | $92.00^{\pm0.2}$ | $91.96^{\pm0.2}$ | $92.22^{\pm0.4}$ | $91.69^{\pm0.4}$ | $97.18^{\pm0.0}$ |

the pseudo labels of the unlabeled data are initialized as negative. For WSC Zhou et al. (2025), unlabeled data are initialized as negative samples, thereby reformulating the PU learning task as a problem of learning with noisy labels. The classification model is subsequently trained using the Learning with Noisy Labels method proposed in WSC.

# F IMPLEMENTATION DETAILS OF *NcPU*

All momentum-related hyperparameters ($\alpha$, $\beta$, and $\gamma$) in *NcPU* are fixed at 0.99 across all datasets. The model is trained for 1300 epochs for all datasets. Optimization is performed using SGD with a momentum coefficient of 0.9, combined with a cosine annealing learning rate scheduler. The initial learning rate is set to 0.01 for the STL-10 dataset, 0.0001 for the xBD dataset, and 0.001 for all other datasets. The $w_r$ is set to 50 for all datasets, while the $w_{ent}$ is assigned a value of 0.5 for CIFAR-100

Table 7: More ablation analysis between $\tilde{\mathcal{L}}_r$ and $s$.

| $\tilde{\mathcal{L}}_r$ | $s$ | CIFAR-10 | | CIFAR-100 | |
|---|---|---|---|---|---|
| | | OA | F1 | OA | F1 |
| | ✓ | $75.61^{\pm1.3}$ | $56.48^{\pm3.4}$ | $61.54^{\pm7.8}$ | $40.58^{\pm22.9}$ |
| ✓ | | $61.60^{\pm0.5}$ | $7.72^{\pm2.4}$ | $50.27^{\pm0.1}$ | $1.09^{\pm0.4}$ |
| ✓ | ✓ | $97.36^{\pm0.1}$ | $96.67^{\pm0.2}$ | $88.28^{\pm0.6}$ | $88.14^{\pm0.9}$ |

Table 8: More ablation analysis on label disambiguation.

| Label Disambiguation | CIFAR-10 | | | | CIFAR-100 | | | |
|---|---|---|---|---|---|---|---|---|
| | OA | F1 | P | R | OA | F1 | P | R |
| $s'$ | $97.41^{\pm0.1}$ | $96.79^{\pm0.1}$ | $95.88^{\pm0.7}$ | $97.72^{\pm0.6}$ | $75.14^{\pm2.7}$ | $79.91^{\pm1.7}$ | $67.15^{\pm2.6}$ | $98.73^{\pm0.5}$ |
| $s' + SAT$ | $60.45^{\pm0.0}$ | $2.24^{\pm0.1}$ | $98.60^{\pm1.2}$ | $1.13^{\pm0.1}$ | $50.25^{\pm0.0}$ | $1.01^{\pm0.1}$ | $97.85^{\pm3.7}$ | $0.51^{\pm0.1}$ |
| $s$ | $97.36^{\pm0.1}$ | $96.67^{\pm0.2}$ | $97.53^{\pm0.4}$ | $95.82^{\pm0.6}$ | $88.28^{\pm0.6}$ | $88.14^{\pm0.9}$ | $89.12^{\pm1.7}$ | $87.27^{\pm3.2}$ |

and 5 for the remaining datasets. A warm-up phase is applied at the beginning of training, during which pseudo targets are kept fixed for the first 30 epochs.

## G  MORE EXPERIMENTAL RESULTS

The results of precision (P), recall (R), and area under receiver operating characteristic curve (AUC) are presented in Table 5-Table 6. *NcPU* achieves a better trade-off between P and R.

## H  MORE ABLATION ANALYSES

$\tilde{\mathcal{L}}_r$ **and $s$ can benefit each other**. Experiments on the CIFAR-10 dataset (Table 7) further validate the theoretical analysis in Section 4, demonstrating that $\tilde{\mathcal{L}}_r$ and $s$ can benefit each other, thereby enabling *NcPU* to achieve superior performance.

**PhantomGate plays an important role in label disambiguation**. More detailed results on label disambiguation are presented in Table 8. While class-conditional prototype-based label disambiguation ($s'$) performs well on the CIFAR-10 dataset, it fails to achieve satisfactory results on the CIFAR-100 dataset. Specifically, this approach achieves relatively high recall for the positive class but suffers from low precision, indicating a tendency to over-identify samples as positive. Therefore, incorporating negative supervision through SAT and PhantomGate is essential, even if it slightly degrades performance on CIFAR-10.

$\tilde{\mathcal{L}}_r$ **significantly enhances the performance of PU learning methods**. Apart from Table 3, we also conducted another set of experiments to analyze the contribution of representation quality on PU learning and test the robustness of $\tilde{\mathcal{L}}_r$ with respect to fewer training data and higher $\pi_p$. CIFAR-100 is used as the experimental dataset, with the division of positive and negative classes consistent with the main experiments, and ResNet-18 serves as the backbone. To highlight the effectiveness of $\tilde{\mathcal{L}}_r$, the $\pi_p$ is set to 0.6, with 1000 positive training samples and 20000 unlabeled training samples. Optimization is performed using SGD with a momentum coefficient of 0.9 and a cosine annealing learning rate scheduler. The initial learning rate is set to 0.001. The $w_r$ is set to 100. The pair-construction strategy is generalized by introducing a similarity threshold ($\tau$), which enables more flexible construction of pairs:

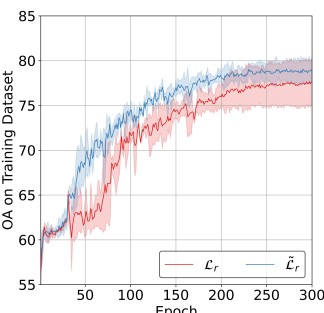

Figure 9: $\tilde{\mathcal{L}}_r$ will accelerate the model's correct fitting to the training dataset.

Table 9: Non-contrastive loss analysis on CIFAR-100 dataset.

| | OA | F1 |
|---|---|---|
| uPU | $66.84^{\pm0.51}$ | $60.12^{\pm1.28}$ |
| nnPU | $69.51^{\pm0.16}$ | $69.52^{\pm0.47}$ |
| uPU+$\mathcal{L}_{self\text{-}r}$ | $72.71^{\pm0.79}$ | $66.80^{\pm1.22}$ |
| uPU+$\mathcal{L}_r$ | $76.29^{\pm3.11}$ | $78.47^{\pm2.48}$ |
| uPU+$\tilde{\mathcal{L}}_r$ | $80.06^{\pm0.03}$ | $79.67^{\pm1.32}$ |
| nnPU+$\tilde{\mathcal{L}}_r$ | $81.55^{\pm0.45}$ | $81.59^{\pm0.61}$ |

$$f_1(\boldsymbol{x}_i)f_1(\boldsymbol{x}_j) + (1 - f_1(\boldsymbol{x}_i))(1 - f_1(\boldsymbol{x}_j)) \geq \tau. \tag{37}$$

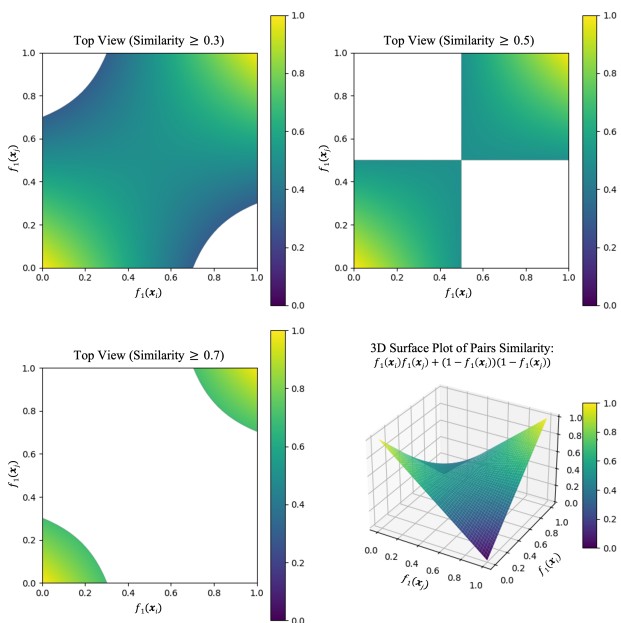

Figure 10: **An illustration of pairs selection based on similarity**. In the top-view representation, the colored regions indicate the pairs involved in training. It can be observed that a smaller $\tau$ leads to a larger number of pairs being included. In particular, when $\tau = 0.5$, the selection becomes equivalent to pairs selection using pseudo labels.

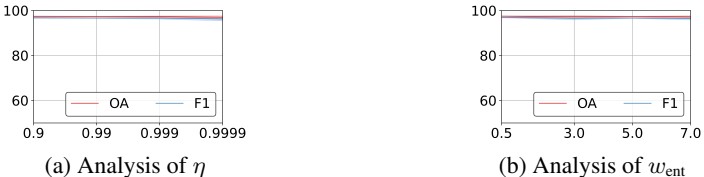

(a) Analysis of $\eta$      (b) Analysis of $w_{\text{ent}}$

Figure 11: Analyses of $\eta$ and $w_{\text{ent}}$.

For example, a lower threshold allows more pairs to participate in training, but also introduces a higher proportion of noisy pairs. As illustrated in the Figure 10, when $\tau = 0.5$, the strategy is equivalent to selecting samples with $\tilde{y}_i = \tilde{y}_j$ as pairs. In this experiment, $\tau$ is set to 0.2 to ensure the inclusion of a sufficient number of pairs during training. As shown in Table 9, $\tilde{\mathcal{L}}_r$ significantly improves the classification performance of the baseline model and accelerates the model's correct fitting of the training data, which may in turn facilitate label disambiguation (Figure 9).

**Analysis on Imbalanced Datasets**. *NcPU* has been validated under more challenging conditions: class imbalance (Table 10) and distribution imbalance (Su et al., 2021; Zhao et al., 2023b) (Table 11). WSC was selected as the comparison method due to its best performance without any auxiliary information. In class imbalance scenarios, the number of positive samples is significantly smaller than that of unlabeled samples. The imbalance ratio (IR) is used to quantify the degree of class imbalance, defined as the ratio of the number of unlabeled samples to positive samples. IR can be increased by reducing the number of positive samples (all other settings remain consistent with the main experiments). Results in Table 10 demonstrate that *NcPU* is robust to class imbalance. Recent studies (Su et al., 2021; Zhao et al., 2023b) have indicated that distribution imbalance is another critical challenge for PU learning tasks: a low $\pi_p$ exerts adverse effects on classifier training. In the experiments of Table 11, 20000 unlabeled samples were used, and distribution imbalance was simulated by decreasing the $\pi_p$ (all other settings are consistent with the main experiments). Results in Table 11 show that *NcPU* is robust to distribution imbalance.

**Analysis of hyperparameters**. $\eta$ denotes the momentum hyperparameter for updating the target network. As shown in Figure 11(a), *NcPU* is robust to $\eta$. In practice, we observe that *NcPU* may

Table 10: Class imbalance results on CIFAR-10 dataset.

| Method | IR: 80 | | IR: 53.33 | |
|---|---|---|---|---|
| | OA | F1 | OA | F1 |
| WSC | $81.46^{\pm 3.0}$ | $71.31^{\pm 6.4}$ | $89.70^{\pm 0.7}$ | $86.59^{\pm 1.4}$ |
| NcPU(ours) | $93.70^{\pm 4.0}$ | $91.34^{\pm 6.1}$ | $96.44^{\pm 0.9}$ | $95.42^{\pm 1.2}$ |

Table 11: Distribution imbalance results on CIFAR-10 dataset.

| Method | $\pi_p$: 0.05 | | $\pi_p$: 0.1 | |
|---|---|---|---|---|
| | OA | F1 | OA | F1 |
| WSC | $90.97^{\pm 0.3}$ | $88.48^{\pm 0.4}$ | $90.72^{\pm 0.3}$ | $88.36^{\pm 0.4}$ |
| NcPU(ours) | $93.57^{\pm 0.5}$ | $91.36^{\pm 0.7}$ | $93.88^{\pm 0.4}$ | $91.81^{\pm 0.6}$ |

Table 12: Computational overhead of different methods on CIFAR-10 dataset.

| Method | Training Phase | Inference Phase | | Task Performance | |
|---|---|---|---|---|---|
| | Training Time/Epoch (s) | Batch Inference (ms/256 samples) | GFLOPs | OA | F1 |
| nnPU | 5.06 | | | $87.29^{\pm 0.5}$ | $83.71^{\pm 0.6}$ |
| LaGAM | 14.79 | | | $95.78^{\pm 0.5}$ | $94.90^{\pm 0.6}$ |
| WSC | 16.00 | 12.92 | 0.56 | $90.55^{\pm 0.3}$ | $87.92^{\pm 0.8}$ |
| NcPU | 14.84 | | | $97.36^{\pm 0.1}$ | $96.67^{\pm 0.2}$ |

occasionally exhibit instability during training with a very small probability, but this issue can be effectively mitigated by the entropy regularization term. Furthermore, the model is also robust to the weighting of the entropy regularization $w_{\text{ent}}$ (Figure 11(b)).

## I   ANALYSIS ON COMPUTATIONAL OVERHEAD

The comparison of computational overhead between *NcPU* and other existing PU learning methods in both training and inference phases is presented in Table 12. Among these methods, nnPU is a classic PU learning algorithm without a representation learning module, while LaGAM, WSC, and NcPU all incorporate representation learning modules: (1) In the training phase, compared with nnPU, the PU learning algorithms with representation learning modules exhibit increased training time per epoch, but there is no significant difference in the per-epoch training time between *NcPU* and the other algorithms with representation learning modules; (2) In the inference phase, *NcPU* and other methods only utilize a single classification network for inference, thus *NcPU* achieves the same inference speed and computational complexity to the other methods; (3) Compared with other methods with representation learning modules, *NcPU* does not show a significant increase in per-epoch training time, yet it delivers a substantial improvement in OA and F1.

## J   THE USE OF LARGE LANGUAGE MODELS

Large Language Models (LLMs) were employed exclusively for the purpose of correcting grammatical errors and enhancing the clarity of expression.

