# OpenReview forum: "Noisy-Pair Robust Representation Alignment for Positive-Unlabeled Learning"
_ICLR.cc/2026/Conference — ICLR 2026 Poster_

### Official Review · Reviewer_TpVx · 2025-10-30

**Soundness:** 2
**Presentation:** 2
**Contribution:** 2
**Rating:** 6
**Confidence:** 3

**Summary:**

This paper addresses Positive-Unlabeled (PU) learning on complex datasets, where existing methods struggle to learn discriminative representations under unreliable supervision. The authors propose NcPU, which combines a noisy-pair robust non-contrastive loss (NoiSNCL) with a phantom label disambiguation (PLD) scheme for conservative negative supervision. The method is theoretically analyzed under the EM framework and empirically validated on complex datasets, showing promising results.

**Strengths:**

1. The t-SNE visualization clearly demonstrates that the proposed method enables the model to learn more discriminative and meaningful features than previous approaches.
2. The authors provide a theoretical analysis to show the effectiveness of their proposed method based on EM framework.
3. Experimental results show that the proposed method surpasses previous methods by a considerable margin.

**Weaknesses:**

1. The novelty of the proposed method is relatively limited. Specifically, it is constructed by incorporating existing techniques such as BYOL, class prototypes, and SAT. The main contribution of the method lies in the introduction of the modified loss term, $\\tilde{\\mathcal{L}}$.
2. The proposed method requires substantially longer training time compared to previous approaches. Specifically, the authors train for 1500 epochs to achieve the best performance, whereas competing methods, such as WSC, are trained for only 250 epochs in their original paper, requiring roughly one-sixth of the training time.

**Questions:**

1. As shown in Equation (7), the authors adopt a square-root loss in the proposed NoiSNCL objective. Consequently, the gradient does not vanish even when the two features of $x_i$ and $x_j$ are identical. Specifically, when $\\tilde{q}_i^\\top \\tilde{q}_j = 1$, the gradient magnitude becomes $\\frac{2}{\\| q_i \\|_2^2}$. Would this non-vanishing gradient potentially lead to overfitting or training instability?

---

> ### Author Response · Authors · 2025-11-20
> **Response to Reviewer TpVx (1/2)**
>
> We sincerely appreciate your constructive suggestions. Below are our responses to the weaknesses and question raised:
>
> **W1**: The novelty of the proposed method is relatively limited. Specifically, it is constructed by incorporating existing techniques such as BYOL, class prototypes, and SAT. The main contribution of the method lies in the introduction of the modified loss term, $\tilde{\mathcal{L}}$.
>
> **A1**: We sincerely appreciate your comment and wish to clarify that there are substantial differences between this work and the existing research:
>
> (1) BYOL aims to address the problem of self-supervised representation learning, however, **the proposed NoiSNCL focuses on noisy-pair robust representation learning**. NoiSNCL successfully extends non-contrastive representation learning from the unsupervised and supervised learning paradigms to the weakly supervised learning (Figure 1);
>
> (2) Existing class prototype techniques (s') only work with auxiliary information (L.237-L.239), yet such auxiliary information is often difficult to estimate accurately (e.g., post-disaster building damage mapping). **The proposed PhantomGate in this paper is the key mechanism for class prototypes to function without any extra information**: PhantomGate allows the model to perform regret-based label updating (s) making the proposed PU learning method applicate to more real-world scenarios. In Table 2, compared with the results of $\tilde{\mathcal{L}}_{\text{r}}$ + s' and $\tilde{\mathcal{L}}$ + s' + SAT, the NcPU achieves improvements of 13.14% and 8.23% in OA and F1, respectively;
>
> (3) Beyond the methodological design, **the proposed NoiSNCL and PLD modules can be theoretically justified to iteratively benefit each other from the perspective of the EM framework** (Section 4);
>
> (4) Compared with other SOTA methods that also do not use any auxiliary information, the proposed NcPU achieves significant OA improvements of 6.81%, 12.89%, and 5.78% on the CIFAR-10, CIFAR-100, and STL-10 datasets, respectively (Table 1).
>
> **W2**: The proposed method requires substantially longer training time compared to previous approaches. Specifically, the authors train for 1500 epochs to achieve the best performance, whereas competing methods, such as WSC, are trained for only 250 epochs in their original paper, requiring roughly one-sixth of the training time.
>
> **A2**: We appreciate your careful observation and would like to clarify the key points as follows:
>
> (1) While the proposed NcPU is trained for 1300 epochs to reach the optimal performance, **it is important to note that NcPU has already achieved an OA of 95.49% on CIFAR-10 and 85.57% on CIFAR-100 at the 250th epoch** (Figure 6), which have significantly outperformed WSC’s corresponding results of 90.55% and 75.39%. This indicates that NcPU delivers superior performance within the same training epoch budget;
>
> (2) As presented in Table 12, the per-epoch training time of NcPU is comparable to other PU learning methods with a representation learning module (e.g., LaGAM, WSC). The extended training epochs are a deliberate choice to demonstrate the stability of the model training and fully exploit the capacity of the proposed NcPU.
>
> Table 12. Computational Overhead of Different Methods on CIFAR-10 Dataset.
> |        | Training Phase          | Inference Phase                   |        | Task Performance |            |
> |--------|-------------------------|-----------------------------------|--------|------------------|------------|
> | Method | Training Time/Epoch (s) | Batch Inference (ms/256 samples)  | GFLOPs | OA               | F1         |
> | nnPU   | 5.06                    | 12.92                             | 0.56   | 87.29±0.5        | 83.71±0.6  |
> | LaGAM  | 14.79                   | 12.92                             | 0.56   | 95.78±0.5        | 94.90±0.6  |
> | WSC    | 16.00                   | 12.92                             | 0.56   | 90.55±0.3        | 87.92±0.8  |
> | NcPU   | 14.84                   | 12.92                             | 0.56   | 97.36±0.1        | 96.67±0.2  |

---

> ### Author Response · Authors · 2025-11-20
> **Response to Reviewer TpVx (2/2)**
>
> **Q1**: As shown in Equation (7), the authors adopt a square-root loss in the proposed NoiSNCL objective. Consequently, the gradient does not vanish even when the two features of $x_i$ and $x_j$ are identical. Specifically, when $\tilde{q} _ {i}^{T}\tilde{q} _ {j}=1$, the gradient magnitude becomes $\frac{2}{||q_i|| _ {2}^{2}}$. Would this non-vanishing gradient potentially lead to overfitting or training instability?
>
> **A3**: Thanks for your comment. Further theoretical and experimental analyses demonstrate that NoiSNCL ($\tilde{\mathcal{L}}_{\text{r}}$) does not introduce significant overfitting or training instability:
>
> (1) As $\tilde{q} _ {i}^{T}\tilde{q} _ {j}$ approaches 1, the gradient magnitude becomes $\frac{2}{||q_i|| _ {2}^{2}}$ instead of infinity. Thus, within a finite number of training iterations, NoiSNCL does not lead to notable overfitting or training instability;
>
> (2) The results of NcPU across different epochs are illustrated in the Figure 6. Taking the results on the CIFAR-10 dataset as an example, NcPU has achieved good performance at the 400th epoch; after prolonged additional training, its results do not exhibit signs of overfitting or instability.

---

> ### Author Response · Authors · 2025-11-28
>
> Dear Reviewer TpVx,
>
> Thank you very much for your thoughtful review and encouraging rating!
>
> As the author–reviewer discussion period is drawing to a close, we would like to kindly follow up to see whether our rebuttal has addressed your concerns. Please let us know if any further clarification would be helpful.
>
> Thank you again for your time and consideration.
>
> Sincerely,
>
> The Authors

---

### Official Review · Reviewer_5HgC · 2025-11-01

**Soundness:** 3
**Presentation:** 3
**Contribution:** 3
**Rating:** 6
**Confidence:** 4

**Summary:**

This paper identifies an insight that noisy pairs tend to dominate the representation learning process, as their gradient magnitudes overwhelm those from the clean pairs. To address this, it proposes a PU learning framework called NcPU. The method introduces two key components: 1) a noisy-pair robust supervised non-contrastive loss (NoiSNCL) that aligns intra-class representations while tolerating noisy pairs through gradient analysis, and 2) a phantom label disambiguation (PLD) module that refines supervision through regret-based label updating. Experimental results demonstrate the framework's effectiveness, with additional results showing its applicability in post-disaster building damage mapping tasks.

**Strengths:**

- This paper provides an insight by identifying and formally analyzing the "noisy-pair gradient dominance" problem in traditional contrastive learning.
- Building on this insight, the proposed method is well-designed. The experimental evaluation is comprehensive, demonstrating state-of-the-art performance across multiple benchmark datasets.
- The paper is generally well-written, explaining the proposed method, with detailed descriptions.

**Weaknesses:**

- The dual-network architecture inevitably increases training cost compared to simpler PU methods. A quantitative comparison of computational overhead would help practitioners evaluate the trade-offs.
- While performance on standard benchmarks is strong, validation under more challenging conditions (e.g., extreme class imbalance, very limited positive samples) would better demonstrate the method's robustness.

**Questions:**

Please refer to the weaknesses.

---

> ### Author Response · Authors · 2025-11-20
> **Response to Reviewer 5HgC**
>
> We sincerely appreciate your thoughtful and constructive feedback. Below are our responses to the weaknesses identified and questions raised:
>
> **W1**: The dual-network architecture inevitably increases training cost compared to simpler PU methods. A quantitative comparison of computational overhead would help practitioners evaluate the trade-offs.
>
> **A1**: Thank you for your insightful suggestion. The comparison of computational overhead between NcPU and other existing PU learning methods in both training and inference phases is presented in Table 12. Among these methods, nnPU is a classic PU learning algorithm without a representation learning module, while LaGAM, WSC, and NcPU all incorporate representation learning modules.
>
> (1) In the training phase, compared with nnPU, the PU learning algorithms with representation learning modules exhibit increased training time per epoch, but **there is no significant difference in the per-epoch training time between NcPU and the other algorithms with representation learning modules**;
>
> (2) In the inference phase, NcPU and other methods only utilize a single classification network for inference, thus **NcPU achieves the same inference speed and computational complexity to the other methods**;
>
> (3) Compared with other methods with representation learning module, NcPU does not show a significant increase in per-epoch training time, yet it delivers a substantial improvement in OA and F1.
>
> Table 12. Computational Overhead of Different Methods on CIFAR-10 Dataset.
> |        | Training Phase          | Inference Phase                   |        | Task Performance |            |
> |--------|-------------------------|-----------------------------------|--------|------------------|------------|
> | Method | Training Time/Epoch (s) | Batch Inference (ms/256 samples)  | GFLOPs | OA               | F1         |
> | nnPU   | 5.06                    | 12.92                             | 0.56   | 87.29±0.5        | 83.71±0.6  |
> | LaGAM  | 14.79                   | 12.92                             | 0.56   | 95.78±0.5        | 94.90±0.6  |
> | WSC    | 16.00                   | 12.92                             | 0.56   | 90.55±0.3        | 87.92±0.8  |
> | NcPU   | 14.84                   | 12.92                             | 0.56   | 97.36±0.1        | 96.67±0.2  |
>
> **W2**: While performance on standard benchmarks is strong, validation under more challenging conditions (e.g., extreme class imbalance, very limited positive samples) would better demonstrate the method's robustness.
>
> **A2**: Thanks for your suggestion. NcPU has been validated under more challenging conditions: class imbalance (Table 10) and distribution imbalance (Table 11). WSC was selected as the comparison method due to its best performance without any auxiliary information.
>
> (1) In class imbalance scenarios, the number of positive samples is significantly smaller than that of unlabeled samples. The imbalance ratio (IR) is used to quantify the degree of class imbalance, defined as the ratio of the number of unlabeled samples to positive samples. IR can be increased by reducing the number of positive samples (all other settings remain consistent with the main experiments). Results in Table 10 demonstrate that NcPU is robust to class imbalance.
>
> Table 10. Class Imbalance Results on CIFAR-10 dataset. IR: Imbalance Ratio
> |        | IR: 80    |           | IR: 53.33 |            |
> |--------|-----------|-----------|-----------|------------|
> | Method | OA        | F1        | OA        | F1         |
> | WSC    | 81.46±3.0 | 71.31±6.4 | 89.70±0.7 | 86.59±1.4  |
> | NcPU   | 93.70±4.0 | 91.34±6.1 | 96.44±0.9 | 95.42±1.2  |
>
> (2) Recent studies have indicated that distribution imbalance is another critical challenge for PU learning tasks: a low $\pi_p$ exerts adverse effects on classifier training. In the experiments of Table 11, 20000 unlabeled samples were used, and distribution imbalance was simulated by decreasing the $\pi_p$ (all other settings are consistent with the main experiments). Results in Table 11 show that NcPU is robust to distribution imbalance.
>
> Table 11. Distribution Imbalance Results on CIFAR-10 dataset.
> |        | $\pi_p$: 0.05 |           | $\pi_p$: 0.1  |            |
> |--------|-----------|-----------|-----------|------------|
> | Method | OA        | F1        | OA        | F1         |
> | WSC    | 90.97±0.3 | 88.48±0.4 | 90.72±0.3 | 88.36±0.4  |
> | NcPU   | 93.57±0.5 | 91.36±0.7 | 93.88±0.4 | 91.81±0.6  |

---

> ### Author Response · Authors · 2025-11-28
>
> Dear Reviewer 5HgC,
>
> Thank you very much for your thoughtful review and encouraging rating!
>
> As the author–reviewer discussion period is drawing to a close, we would like to kindly follow up to see whether our rebuttal has addressed your concerns. Please let us know if any further clarification would be helpful.
>
> Thank you again for your time and consideration.
>
> Sincerely,
>
> The Authors

---

### Official Review · Reviewer_PR5m · 2025-11-03

**Soundness:** 4
**Presentation:** 3
**Contribution:** 3
**Rating:** 6
**Confidence:** 4

**Summary:**

This paper introduces NcPU, a non-contrastive PU learning framework that addresses the representation separability in PU learning. The proposed method integrates a noisy-pair robust non-contrastive loss (NoiSNCL) and a phantom label disambiguation (PLD) module. NoiSNCL enhances intra-class representation alignment under noisy supervision, while PLD refines pseudo labels through prototype-based, regret-aware updates. Theoretically, the framework is grounded in an EM-based interpretation, and empirically, NcPU achieves substantial gains across multiple datasets, even surpassing supervised baselines.

**Strengths:**

1. The paper clearly identifies the representation separability issue as the bottleneck in PU learning, and the motivation is well justified and convincing.
2. The proposed noisy-pair robust loss effectively mitigates the influence of false labels, a common problem in PU learning under weak supervision.
3. The proposed method achieves significant and consistent improvements, even outperforming supervised models on several benchmarks.
4. The paper is well-organized and clearly written, making the technical ideas easy to follow.

**Weaknesses:**

1. The loss function  $\tilde{\mathcal{L}}_r = 2\sqrt{1 - \langle q, k \rangle}$may cause numerical instability when $\langle q, k \rangle \approx 0$, since the gradient involves a term  $\frac{1}{\sqrt{1 - \langle q, k \rangle}} \to \infty$. Although the composed gradient may remain finite, the authors should discuss the numerical stability and possible mitigation strategies.

2. The idea of introducing robust non-contrastive alignment has conceptual overlap with prior work such as the CoTAP loss proposed in “Semantic Concentration for Self-Supervised Dense Representations Learning” (TPAMI 2025).
   While the problem settings differ, it would strengthen the contribution to discuss distinctions or novel theoretical insights explicitly.

3. A naïve self-supervised baseline (e.g., BYOL or DINO pretrained representations combined with standard PU learning) is missing, which would help isolate the contribution of NoiSNCL and PLD from pretrained feature quality.

4. The paper still uses BYOL as its base self-supervised method, which has been surpassed by recent approaches like iBOT (ICLR 2022) and DINO v1/v2.  Future work could benefit from integrating NcPU with stronger backbone frameworks to further validate its robustness.

5. The proposed framework involves multi-view alignment and prototype-based refinement, which may increase training time and computational cost.  A runtime comparison with existing PU learning or non-contrastive baselines would clarify efficiency and practical feasibility.

Overall, this paper is worthy of acceptance. Once the above issues are adequately addressed, I would be inclined to raise my score.

**Questions:**

Please see the weaknesses.

---

> ### Author Response · Authors · 2025-11-20
> **Response to Reviewer PR5m (1/2)**
>
> We sincerely appreciate your valuable suggestions and insightful comments. We address the specific questions and concerns raised as follows:
>
> **W1**: The loss function $\tilde{\mathcal{L}}_{\text{r}}=2 \sqrt{1-\langle q,k \rangle}$ may cause numerical instability when $\langle q,k \rangle \approx 0$, since the gradient involves a term $\frac{1}{\sqrt{1-\langle q,k \rangle}} \rightarrow \infty$. Although the composed gradient may remain finite, the authors should discuss the numerical stability and possible mitigation strategies.
>
> **A1**: Thanks for your suggestion. The discussion on numerical stability and potential mitigation strategies are as follows:
>
> (1) Benefiting from the asymmetric architecture (the online network includes a prediction head, while the encoder and projection head in the target network adopt the EMA parameters of their corresponding structures in the online network) and random data augmentation, $q \neq k$ is ensured, which can enable the numerical stability of the gradient;
>
> (2) The training stability can also be empirically verified (Figure 6, P. 9): Taking the results on the CIFAR-10 dataset as an example, NcPU has achieved promising performance at the 400th epoch. After extended training for a longer period, the results of NcPU does not exhibit overfitting or instability;
>
> (3) The numerical stability can be improved by constraining the value of $\langle q,k \rangle$ (e.g., $\lbrack 10^{-4},1-10^{-4} \rbrack$).
>
> **W2**: The idea of introducing robust non-contrastive alignment has conceptual overlap with prior work such as the CoTAP loss proposed in “Semantic Concentration for Self-Supervised Dense Representations Learning” (TPAMI 2025). While the problem settings differ, it would strengthen the contribution to discuss distinctions or novel theoretical insights explicitly.
>
> **A2**: Thank you for your suggestion. CoTAP concentrates on the self-supervised dense representation learning task, however, this paper focuses on PU learning task. Regarding the tolerance of noisy pairs, the negative effect of noises is alleviated by assigning higher weights to sample pairs with top scores in CoTAP Loss. In contrast, NoiSNCL tolerates noisy pairs from the perspective of gradients. The comparison regarding the CoTAP Loss has been updated as follows: “The CoTAP Loss is proposed to align the representations of semantically similar objects in self-supervised dense representation learning, and the negative effect of noises is alleviated by assigning higher weights to sample pairs with top scores”. (P. 10)
>
> **W3**: A naïve self-supervised baseline (e.g., BYOL or DINO pretrained representations combined with standard PU learning) is missing, which would help isolate the contribution of NoiSNCL and PLD from pretrained feature quality.
>
> **A3**: Sincere thanks for your suggestion. The contribution of representation quality on standard PU learning method (uPU) is demonstrated in the following table (Table 9). Performance of the naïve self-supervised baseline (BYOL + standard PU learning method=$\mathcal{L} _ {\text{self-r}}$+uPU, OA:72.71, F1:66.80) is outperformed by $\tilde{\mathcal{L}}_{\text{r}}$+uPU (OA: 80.06, F1:79.67).
>
> |          | OA         | F1         |
> |------------|------------|------------|
> | uPU| 66.84±0.51| 60.12±1.28 |
> | uPU+BYOL | 72.71±0.79 | 66.80±1.22 |
> | uPU+$\mathcal{L}_{\text{r}}$| 76.29±3.11 | 78.47±2.48 |
> | uPU+$\tilde{\mathcal{L}}_{\text{r}}$| 80.06±0.03 | 79.67±1.32 |
>
> **W4**: The paper still uses BYOL as its base self-supervised method, which has been surpassed by recent approaches like iBOT (ICLR 2022) and DINO v1/v2. Future work could benefit from integrating NcPU with stronger backbone frameworks to further validate its robustness.
>
> **A4**: Thanks for your constructive suggestion. This paper endeavors to develop a PU classifier that achieves both improved performance and strong theoretical interpretability. While recent self-supervised representation learning methods (iBOT and DINO v2) integrating masked image modeling (MIM) with contrastive/non-contrastive learning have achieved better representation learning results, the inherent complexity of combining MIM with contrastive/non-contrastive learning poses significant challenges to theoretical analysis of the model. In contrast, the BYOL framework, which relies on non-contrastive learning, employs only $\mathcal{L} _ {\text{self-r}}$ for representation learning. This simplicity enables NcPU to be theoretically explained from the perspective of the EM framework. In future work, we will attempt to integrate MIM into NcPU, aiming to utilize more powerful backbone frameworks while providing corresponding theoretical interpretations. The updated future work is as follows: “Masked image modeling will be incorporated to harness more powerful backbone frameworks”. (P. 10)

---

> ### Author Response · Authors · 2025-11-20
> **Response to Reviewer PR5m (2/2)**
>
> **W5**: The proposed framework involves multi-view alignment and prototype-based refinement, which may increase training time and computational cost. A runtime comparison with existing PU learning or non-contrastive baselines would clarify efficiency and practical feasibility.
>
> **A5**: Thank you for your insightful comment. The comparison of computational overhead between NcPU and other existing PU learning methods in both training and inference phases is presented in Table 12. Among these methods, nnPU is a classic PU learning algorithm without a representation learning module, while LaGAM, WSC, and NcPU all incorporate representation learning modules.
>
> (1) In the training phase, compared with nnPU, the PU learning algorithms with representation learning modules exhibit increased training time per epoch, but **there is no significant difference in the per-epoch training time between NcPU and the other algorithms with representation learning modules**;
>
> (2) In the inference phase, NcPU and other methods only utilize a single classification network for inference, thus **NcPU achieves the same inference speed and computational complexity to the other methods**;
>
> (3) Compared with other methods with representation learning module, NcPU does not show a significant increase in per-epoch training time, yet it delivers a substantial improvement in OA and F1.
>
> Table 12. Computational Overhead of Different Methods on CIFAR-10 Dataset.
> |        | Training Phase          | Inference Phase                   |        | Task Performance |            |
> |--------|-------------------------|-----------------------------------|--------|------------------|------------|
> | Method | Training Time/Epoch (s) | Batch Inference (ms/256 samples)  | GFLOPs | OA               | F1         |
> | nnPU   | 5.06                    | 12.92                             | 0.56   | 87.29±0.5        | 83.71±0.6  |
> | LaGAM  | 14.79                   | 12.92                             | 0.56   | 95.78±0.5        | 94.90±0.6  |
> | WSC    | 16.00                   | 12.92                             | 0.56   | 90.55±0.3        | 87.92±0.8  |
> | NcPU   | 14.84                   | 12.92                             | 0.56   | 97.36±0.1        | 96.67±0.2  |

---

> > ### Comment · Reviewer_PR5m · 2025-11-24
> >
> > Thanks for your detailed responses. The concerns are well-addressed. For the observation that the baseline (uPU + BYOL) performs poorly (A3), I suggest providing a deeper analysis of the underlying causes in the latest version, rather than only presenting performance comparisons.

---

### Author Response · Authors · 2025-12-01
**Rebuttal Summary by the Authors**

Dear PCs, SACs, ACs, and Reviewers,

Thank you very much for your time and effort in reviewing our submission. To assist the newly assigned AC and help reduce their workload, we provide below a summary of the key points from the reviews and the reviewer-author discussions.

---

**Strength.** Overall, we are encouraged that the reviewers gave this paper a highly positive evaluation in the initial reviews.  Specifically:

**1.The problem of noisy-pair robust representation learning is clearly defined (PR5m, 5HgC):** this issue is a common challenge under unreliable supervision, not only in PU learning but also potentially in broader weakly supervised learning settings.

**2.The proposed NcPU framework is well designed (PR5m, 5HgC, TpVx):** It addresses the above problem from a novel gradient-based perspective.

**3.Solid theoretical guarantees (PR5m, TpVx):** The proposed NcPU framework is theoretically grounded and can be interpreted under the EM framework.

**4.Comprehensive evaluation (PR5m, 5HgC, TpVx):** The proposed method consistently surpasses prior approaches by a considerable margin on multiple benchmarks, and even outperforms supervised methods.

**5.Clear writing (PR5m, 5HgC):** The manuscript is well-structured and clearly presented.

---

**Concerns and Our Addressing.** During the discussion period, we actively addressed each of the reviewers’ comments point by point, which were recognized by the reviewer and led to increased scores. The main updates are summarized as follows:

**1.Computational overhead comparison (PR5m, 5HgC, TpVx):** We demonstrate that the proposed method incurs no significant training or inference overhead compared with other approaches that also incorporate a representation-learning module (Table 12).

**2.Training stability analysis (PR5m, TpVx):** Both the theoretical analysis and empirical results confirm that NoiSNCL does not introduce overfitting or training instability (Figure 6).

**3.Additional experiments (PR5m, 5HgC):** 1) BYOL combined with standard PU method further highlights the contribution of representation quality (Table 9); 2) Experiments on imbalanced datasets demonstrate the robustness of the proposed method (Table 10, Table 11).

**4.Clarification of novelty (TpVx):** Unlike existing self-supervised and supervised representation learning methods, this paper focuses on noisy-pair robust representation learning (Figure 1).

**5.Improved related work and future work sections (PR5m):** The reviewers’ suggestions have been incorporated to improve the related work and future work sections.

---

**Recognition of Our Revision from Reviewers.** During the discussion period, reviewer PR5m explicitly acknowledged that the concerns had been well addressed and raised the score (6→8). We believe that we have also properly addressed the concerns of the remaining two reviewers.

---

Thank you again for your time and consideration.

Sincerely,

The Authors

---

### Meta-Review · Area_Chair_DqPP · 2026-01-04

**Summary:**

This work addresses the key performance gap between PU learning and supervised learning on complex datasets by identifying representation learning under unreliable supervision as a central bottleneck. The authors propose NcPU, combining (i) NoiSNCL, a noisy‑pair robust supervised non‑contrastive loss motivated by a gradient‑dominance analysis, and (ii) PLD/PhantomGate, which is a negative‑supervision mechanism with regret‑based pseudo‑label updating that does not require auxiliary negatives or a pre‑estimated class prior.
Across three reviews, strengths were consistent which has helped to reach this decision, i.e. clear problem motivation, coherent method design, and strong empirical results across multiple benchmarks and a post‑disaster building damage mapping application.

Reviewers’ main concerns were on (a) training/numerical stability of the square‑root style loss, (b) novelty vs. combining known components (BYOL/prototypes/SAT), (c) the need for a naive SSL+PU baseline to isolate representation quality benefits, and (d) computational/epoch budget fairness.

I believe that after rebuttal and revisions, and the authors detialed responses to the reviewers' concerns, the stability and efficiency concerns are substantially addressed with explicit analysis and added experiments (including stability curves and overhead table), and the baseline request is partially addressed via added non‑contrastive baseline comparisons; one reviewer (PR5m) managed to respond before the change of process took place, explicitly indicating that their concerns were well‑addressed and requested only deeper causal analysis for the weak SSL baseline.

Overall, the method is technically sound, clearly motivated, and empirically strong, with the remaining issues primarily in the depth of presentation and analysis rather than correctness.

**Reviewer Concerns:**

I do not think the reviewers had major concerns in the first place hence their scores. But in summary the issues were as follows:

a) Numerical stability / non‑vanishing gradients in NoiSNCL

This was raised by PR5m and TpVx (stability/overfitting risk due to square‑root loss and gradient terms).
This was addressed by the authors nicely citing the paper and some additional information.

b) Novelty: “composition of BYOL + prototypes + SAT; only loss is new”

This was raised by TpVx and PR5m, with the latter also requesting a clearer distinction vs related work (e.g. CoTAP).
I think the authors responded satisfactorily again. Nevertheless, I think that the paper could still benefit from even clearer contributions to be clear what the novel components are.

c) Baseline isolation/ablations: need naive SSL+PU baseline (e.g. BYOL/DINO + uPU)

this was raised by PR5m. In my view this was partially addressed. The requested naive baseline comparison is now present, but the causal analysis (e.g. mismatch between SSL objective and PU label noise dynamics) could be expanded in the final version, hopefully in the camera ready version.

d) Computational cost and training‑epoch fairness (e.g. 1300/1500 epochs vs baselines trained fewer epochs)

All reviewed picked upon this.
The rebuttal argues NcPU already surpasses key baselines at earlier epochs (via OA curves), and long training is partly to demonstrate stability and extract best performance.
My view is that this is addressed. However, for fairness/clarity, the camera‑ready could add a compact table reporting performance at a standardised epoch run for key methods in addition to the 1300‑epoch headline.

**Reviewer Scores:**

I believe that it is likely that all reviewers would have raised their scores.

Reviewer PR5m explicitly states the concerns are well addressed in the discussion; the remaining request is only a deeper analysis of one baseline behaviour, which is non‑blocking. So this one is rather straightforward.

Reviewer 5HgC's primary concerns (overhead; robustness under challenging conditions) were directly addressed with new Table 12 and imbalance experiments (Tables 10–11); likely to have increased their score with further discussion.


Reviewer TpVx's novelty and long‑training critiques are partially mitigated by more apparent gradient‑based motivation, stability analysis, and early‑epoch performance discussion. However this reviewer's review was very brief and confidence was fair, so it is likely they would either disengage or leave their score unchanged.

---

### Decision · Program_Chairs · 2026-01-26

Accept (Poster)